


# A Bayesian network approach to modelling rip current drownings and shore-break wave injuries

Elias de Korte[1,2], Bruno Castelle[3,4], and Eric Tellier[5,6,7]

[1]Utrecht University, Faculty of Geosciences, Utrecht, The Netherlands
[2]TU Delft, Delft, The Netherlands
[3]CNRS, UMR EPOC, Univ. Bordeaux, Pessac, France
[4]Univ. Bordeaux, UMR EPOC, CNRS, Pessac, France
[5]INSERM, ISPED, Centre INSERM U1219 Bordeaux Population Health Research, Univ. Bordeaux, Bordeaux, France
[6]ISPED, Centre INSERM U1219 Bordeaux Population Health Research, Univ. Bordeaux, Bordeaux, France
[7]Pôle Urgences Adultes, CHU de Bordeaux, SAMU-SMUR, Bordeaux, France

**Correspondence:** Bruno Castelle (bruno.castelle@u-bordeaux.fr)

**Abstract.** A Bayesian network (BN) approach is used to model and predict shore-break related injuries and rip-current drowning incidents based on detailed environmental conditions (wave, tide, weather, beach morphology) on the high-energy Gironde coast, southwest France. Six years (2011-2017) of boreal summer (June 15 - September 15) surf zone injuries (SZIs) were analysed, comprising 442 (fatal and non-fatal) drownings caused by rip currents and 715 injuries caused by shore-break waves.

Environmental conditions at the time of the SZIs were used to train two separate Bayesian networks (BNs), one for rip current drownings and the other one for shore-break wave injuries, each one with a hidden hazard and exposure variables. Both BNs were tested for varying complexity using $K$-fold cross-validation based on multiple performance metrics. Validation (prediction) results slightly improve predictions of SZIs with a poor to fair skill based on a combination of different metrics. Shore-break related injuries appear more predictable than rip current drowning incidents as the shore-break BN systematically

performed better than the rip current BN. Sensitivity and scenario analyses were performed to address the influence of environmental data variables and their interactions on exposure, hazard and resulting life risk. Most of our findings are in line with earlier SZI and physical hazard-based work, that is, that more SZIs are observed for warm sunny days with light winds, long-period waves, with specifically more shore-break related injuries at high tide and for steep beach profiles, and more rip current drownings near low tide with near shore-normal wave incidence and strongly alongshore non-uniform surf zone morphology.

The BNs also provided fresh insight, showing that rip current drowning risk is approximately equally distributed between exposure (variance reduction $Vr = 14.4\%$) and hazard ($Vr = 17.4\%$), while exposure of water user to shore-break waves is much more important ($Vr = 23.5\%$) than the hazard ($Vr = 10.9\%$). Large surf is found to decrease beachgoer exposure to shore-break hazard, while this is not observed for rip currents. Rapid change in tide elevation during days with large tidal range was also found to result in more drowning incidents, presumably because it favors the rapid onset of rip current activity and can therefore surprise unsuspecting bathers. We advocate that such BNs, providing a better understanding of hazard, exposure

and life risk, can be developed to improve public safety awareness campaigns, in parallel with the development of more skillful risk predictors to anticipate high life-risk days.



## 1 Introduction

Wave-dominated beaches offer a playground for a variety of activities, but at the same time they pose a threat to water users.
There are two primary causes of surf zone injuries (SZIs), which can sometimes co-exit at the same beach (Castelle et al., 2018):
i) rip currents resulting in drowning incidents and ii) shore-break waves which can result in e.g. spine and shoulder dislocations.
Rip currents are intense seaward-flowing narrow currents which can form through different driving mechanisms related to
breaking waves (Dalrymple et al., 2011; Castelle et al., 2018). They form close to the shoreline and often extend beyond the
surf zone. Therefore they can transport unsuspecting bathers offshore, who potentially panic and drown (Drozdzewski et al.,
2012; Brighton et al., 2013). The shore-break wave hazard has received little attention in the literature compared to rip current
hazard. However, shore-break waves can cause a large amount of injuries (Puleo et al., 2016), including severe spine injuries
(Robbles, 2006). At certain beaches, shore-break waves can even be the primary cause of SZIs, e.g. up to 88% at Ocean City,
Maryland (Muller, 2018).

Rip flow speed, which is a proxy of rip current hazard, has been addressed on many beaches through both field measurements
and numerical modelling (see Castelle et al., 2016, for a review). In brief, rip flow speed generally increases with increasing
wave height and period (e.g. MacMahan et al., 2006), more shore-normal incidence (e.g. MacMahan et al., 2005), generally
lower tide levels (e.g. Brander and Short, 2001; Austin et al., 2010; Bruneau et al., 2011; Houser et al., 2013) and more
alongshore-variable surf zone morphology (Moulton et al., 2017). It is also well known that shore-break waves are associated
with steep beaches and longer period waves (Battjes, 1974; Balsillie, 1985). In addition, the number of SZIs is also greatly
influenced by the number of beachgoers exposing themselves to surf zone hazards. Given that warm sunny days with low winds
typically result in increased beach attendance (Ibarra, 2011; Dwight et al., 2007), it is expected that during such days the life
risk, and thus the number of SZIs, are increased.

Prominent environmental controls on SZIs were identified by comparing the frequency distribution of an environmental
variable (e.g. significant wave height $H_s$, tide elevation $\eta$) during an injury, with the background frequency distribution of that
variable (Scott et al., 2014; Castelle et al., 2019). The difference between two frequency distributions shows the disproportion-
ate amount of conditions that are associated with SZIs. At two different beaches along the Atlantic coast of Europe, Scott et al.
(2014) and Castelle et al. (2019) showed that the number of drowning incidents occur during warm sunny days with light wind,
maximizing beach attendance, and shore-normally incident long-period waves, maximizing rip current activity. Although such
analysis provides an indication of the prominent environmental controls, it does not uncover the interplay between variables,
and the relative magnitude of each variable. A related challenge based on current research is filtering the effect of how water
users choices are influenced by environmental conditions (e.g. wave height $H_s$). For instance, it is expected that high surf and
heavy shore-break waves discourage an amount of the beachgoers from entering the water, even on warm sunny days, resulting
in less exposure. Finally, the respective contributions of hazard and exposure to the overall life risk for shore-break waves and
rip current are virtually unknown.

Prediction of SZIs together with a better understanding of the interplay between weather and marine conditions and effect
on life risk at the beach, could help a better anticipation of high risk, mass rescue days and further improve public safety




awareness campaigns on surf zone hazards. This requires a high order statistical approach like a Bayesian Network (BN). BNs are probabilistic graphical models that are based on a joint probability distribution of a set of variables with a possible mutual causal relationship. BNs have been previously successfully used in coastal science, estimating morphological changes, changes

in wave parameters in the surf-zone and coastal flood risks (Gutierrez et al., 2011; Plant and Holland, 2011; Fienen et al., 2013; Pearson et al., 2017). Stokes et al. (2017) compared a BN to a multiple linear regression approach to model exposure, hazard and, in turn, life risk to beach users at 113 lifeguarded beaches in UK. Even though the multiple linear regression method moderately outperformed the BN, Stokes et al. (2017) acknowledged the benefits of a BN approach to identify of the characteristics of high risk beaches from a large data set. More recently, Doelp et al. (2019) used a BN to predict SZIs on the

Delaware Coast, which are primarily caused by shore-break waves (Puleo et al., 2016). They showed that a BN approach can improve predictions 69.7 % of the time, but also acknowledged limitations in predicting anomalous injuries. A BN approach has the potential both to show good prediction skill to assist decision-making and to provide a better understanding of rip current and shore-break hazards.

In this paper, a data set (2011-2017) of 442 drowning injuries (fatal and non-fatal) and 715 shore-break injuries occurring in

boreal summer (June 15 - September 15) and corresponding environmental conditions along the Gironde coast in south-west France are used to create BNs for rip current related drownings and shore-break injuries. The study area and SZI dataset are described in Section 2. Section 3 presents the development of the BNs and the method used to train them and address their performance. Results are shown in Section 4 and are further discussed in Section 5 before conclusions are drawn.

## 2   Environmental and SZI dataset along the Gironde Coast

### 2.1   Study area

The Gironde coast is located in southwest of France and stretches approximately 140 km from the La Salie Beach (La Teste) in the south to the Gironde Estuary in the north, and is interrupted by the Arcachon tidal inlet (Figure 1a). It is a meso-macrotidal environment with spring tidal range reaching 5 m. Wave conditions vary seasonally with a 99.5% exceedance significant wave height $H_s$ of 5.6 m, and occasional severe storms with $H_s > 8$ m. Summers are associated with smaller waves with a mean $H_s$

of around 1.2 m and a dominant W-NW incidence (Castelle et al., 2019).

The coast is composed of high-energy sandy beaches backed by high and wide coastal dunes. Beaches are intermediate double barred, with deep and more or less regular rip channels incising the intertidal inner bar with an average spacing of approximately 400 m (Figure 1a and b). Intense rip currents can flow through the rip channels, with rip flow intensity potentially exceeding 1 m/s even for low-energy ($H_s < 1$ m) long-period waves (Castelle et al., 2016). Rip current flow is strongly

modulated by the tide level, with maximum rip current activity occurring between low and mid tide in typical summer wave conditions (Bruneau et al., 2011). In winter, more energetic wave conditions drive a more dissipative gently sloping beach face. In contrast, the upper part of the beach is steeper in summer due to smaller waves. Beach slope and rip channel morphology also show a large interannual variability enforced by large interannual variability in the wave climate (Dodet et al., 2019). Overall, beach states are similar along the coast but with increasingly steep beach face, deeper and more spaced rip channels



southwards. Noteworthy, beach morphology dramatically changes along sectors adjacent to the Arcachon lagoon and Gironde estuary where rip current activity decreases, but tide-driven currents become substantial (> 0.2 m/s during ebb and flood).

The Gironde coast is known for a large population of tourists visiting the beaches, which results in large amounts of injuries sustained by beachgoers and surfers of all levels (Figure 1c) (Castelle et al., 2018; Tellier et al., 2019). Beaches are patrolled by lifeguards during the summer months of July and August. Patrolled periods are extended approximately from the 15th of
June to the 15th of September at the busiest beaches. A designated and supervised bathing zone is delimited by two blue flags. However, many remote beach access paths through coastal dune tracks and many access points are situated on unpatrolled sections of beaches, kilometres away from any lifeguard presence (Castelle et al., 2019).

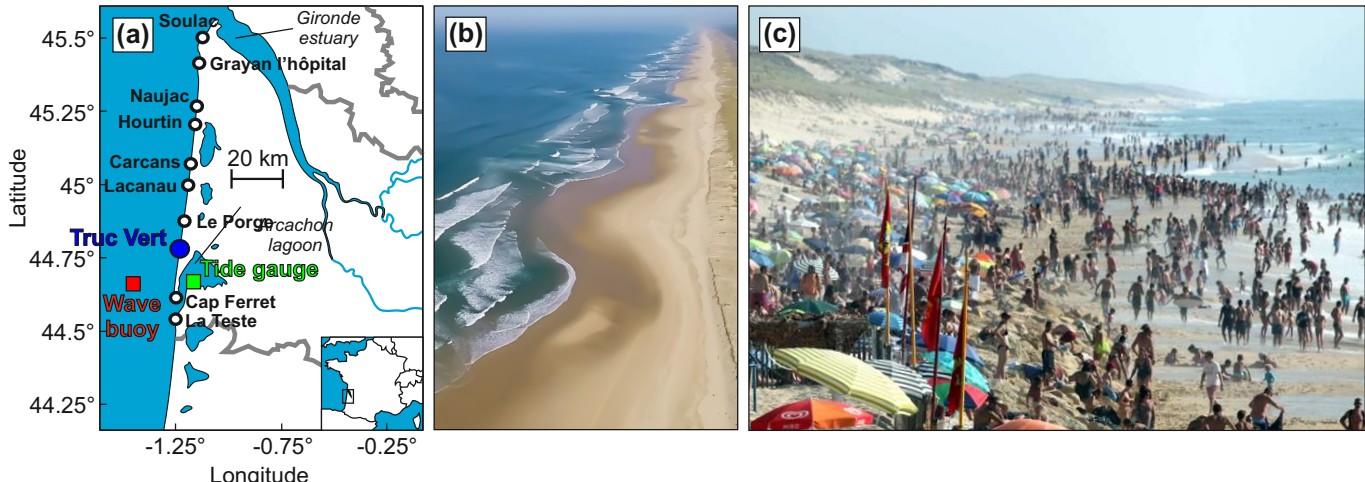

**Figure 1.** (a) Location map of the Gironde Coast, southwest France. Black circles indicate municipalities where injuries were reported. Locations of Truc Vert beach and wave and tide data used in this study are also indicated; (b) aerial photograph of Truc Vert beach at low tide, exposing rip channels (Ph. V. Marieu); (c) crowded Lacanau beach at summer during a high tide (Ph. J. Lestage).

### 2.1.1  SZI data

The SZI dataset used herein is detailed in Tellier et al. (in revision, i). In short, SZIs were recorded by the medical emergency
call center SAMU (Service d'Aide Médicale d'Urgence) of Bordeaux for the Gironde department. Calls from beachgoers and lifeguards dealing with drowning or rescues received between January 2011 and November 2017 were used here. Excluding training calls, duplicates and calls lacking victims, a total of 5022 injuries were collected. Table 2 shows that the discrepancy between the total number of injuries and the combined shore-break and rip current related SZIs is due to the large number of calls with insufficient information collected. Noteworthy, the 916 surfing-related injuries (Table 1) occurring during this period
were also disregarded for analysis. The reason for this being that a large number of surfer injuries involve collision with other surfers and are likely influenced by other factors (e.g. surf break quality, surf school activity) which are not related to physical hazards.



A SZI was classified as shore-break when the medical file stated explicitly: "shore-break" or a French equivalent. Given that along this coast approximately 80% of the drowning incidents involving bathers are caused by rip currents (Castelle et al.,
2018), rip-related SZIs (drownings) were determined if a drowning stage (according to standardized medical classification) was reported in the medical file, with two notable exceptions. Drownings that were related to shore-break waves were classified as shore-break, because they were presumably not associated with rip currents. Similarly, surfing related drownings were excluded as there is no evidence that most of the drownings of surfers are related to rip currents. Six mutually exclusive classes were found based on activity (see Table 2). Even though the activity was unknown for 1943 of the SZIs, drowning stage and the
shore-break classifier provided the information to classify some of these SZIs as shore-break or rip related drowning. Amongst the population, 45% was male, 33% female and for 21% the sex was not recorded. The population is relatively young with 43% between 6 and 19 years old. A slightly elevated number of SZIs was found for the age group between 36-45 years old (see Figure 2). This demographics is in line with another, shorter, dataset described in Castelle et al. (2018). By far most SZIs occurred at Lacanau beaches (26%), which is one of the most popular beaches that consequently attracts crowds in summer
(see Figure 2).

**Table 1.** Activity distribution SZIs as indicated on the medical files between 08-Jan-2011 and 18-Nov-2017

| Activity | nb SZIs |
|---|---|
| swimming | 1229 |
| surfing | 827 |
| body-boarding | 89 |
| beach-related | 898 |
| skim-boarding | 36 |
| unknown activity | 1943 |
| **total** | **5022** |

**Table 2.** Post-processed categories to distinguish between rip related drownings and shore-break injuries

| Class | nb SZIs |
|---|---|
| swimming (non-drowning) | 282 |
| **rip related drowning** | **575** |
| **shore-break** | **750** |
| surfing/body-boarding | 916 |
| beach-related | 934 |
| unknown | 1565 |
| **total** | **5022** |

For the purpose of this study, only summer periods between June 15 and September 15 were taken for each year because outside of this period SZIs become extremely rare events, which poses challenges for BN training. In the summer periods 442 drowning SZIs and 715 shore-break SZIs where found. This is the final population that was used in the Bayesian network.

## 2.2 Environmental data

Environmental conditions were estimated at the time of each SZI by using a data set comprising tide, wave and weather data. The dataset is described in detail in Castelle et al. (2019). Hourly weather data was collected at the Météo-France weather station Cap Ferret (Figure 1a) from the RADOME (Réseau d'Acquisition de Données d'Observations Météorologiques Etendu). A tidal component analysis of a 3-month time series of continuous, storm-free, Eyrac tide gauge data (Fig. 1) was performed to reconstruct a tide level time series at 10-min interval. The average phase lag between the Eyrac tide gauge and beaches of

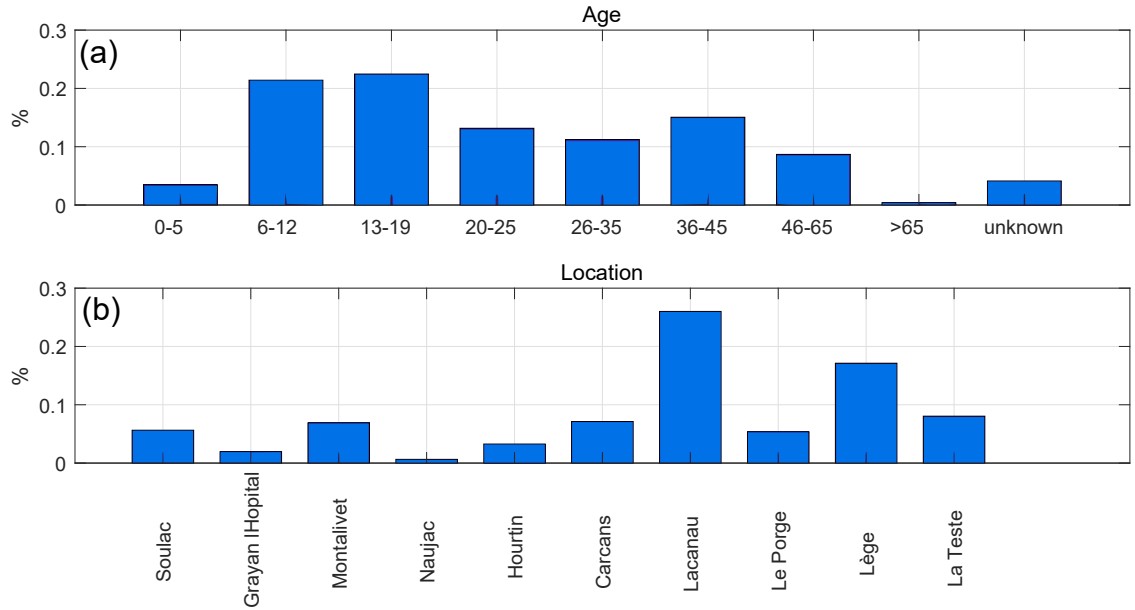

**Figure 2.** Distribution of (a) age and (b) location of SZIs between 08-Jan-2011 and 18-Nov 2017. Beaches are ordered from North (left) to South (right)

the study area was estimated using tide charts from the Service Hydrographique et Océanographique de la Marine (France), resulting in a maximum tide elevation error of 0.3 m at all sites. A wave model hindcast was used to provide continuous wave conditions at the coast. The WaveWatch 3 (Tolman, 2014) hindcast was performed on an unstructured grid with a resolution increasing from 10 km offshore to 200 m near the coast (Boudière et al., 2013). Wave conditions were extracted at an in situ directional wave buoy location c. 10 km offshore of Truc Vert at ca. 50-m depth, and have been extensively validated with

field data (e.g. Castelle et al., 2020). The primary metocean variables used are: tidal elevation ($\eta$), significant wave height ($H_s$), mean wave period ($T_{02}$), wave direction ($\theta$), temperature ($T$), wind speed ($U$), Insolation ($I$). From tidal elevation, tidal range ($TR$) and tidal gradient ($d\eta$) where derived. Maximum, minimum, mean and standard deviation summer statistics are summarized in Table 3.

Previous work along this stretch of coast showed, qualitatively, the importance of beach upper beach slope and rip channel

development on shore-break related injuries and drowning incidents, respectively (Castelle et al., 2019). To further quantitatively address this link with the longer dataset used herein, we used monthly to bimonthly topographic surveys performed at Truc Vert beach since 2003 (the reader is referred to Castelle et al., 2020, for a detailed description of this beach monitoring program). This dataset was used to derive two morphological metrics. First, the inverse foreshore slope ($IFS$) was calculated





**Table 3.** Statistics of environmental conditions of SZI records during summers between June 15 - September 15

| Environmental variable | maximum | minimum | mean | standard deviation |
|---|---|---|---|---|
| Significant wave height $H_s$ (m) | 3.13 | 0.23 | 1.17 | 0.38 |
| Mean wave period $T_{02}$ (s) | 11.64 | 2.66 | 6.19 | 1.94 |
| Wave direction $\theta$ (°) | 340.10 | 247.30 | 291.67 | 7.82 |
| Tidal elevation $\eta$ (m) | 2.21 | -2.26 | -0.03 | 1.09 |
| Tidal range $TR$ (m) | 4.52 | 1.64 | 3.12 | 0.69 |
| Tidal gradient $d\eta$ (m H$^{-1}$) | 0.51 | -0.51 | 0.05 | 0.26 |
| Temperature $T$ ($C°$) | 36.23 | 15.47 | 25.07 | 3.36 |
| Insolation $I$ (min H$^{-1}$) | 60 | 0 | 48.17 | 14.77 |
| Wind speed $U$ (m s$^{-1}$) | 12.65 | 1.2 | 4.56 | 1.26 |

as:

$$IFS = \frac{1}{\tan(\beta)} \qquad (1)$$

where $\tan(\beta)$ is the beach slope between 1 m and 3 m above mean sea level (amsl) from a linear regression. To filter out extreme alongshore variations in $IFS$, the slope was averaged over four cross-shore transects that were systematically surveyed during the monitoring program (Figure ).

Sinuosity ($S$) of the mean sea level iso-contour line was used to provide a measure rip channel development. It was defined as:

$$S = \frac{L_t}{L_s} \qquad (2)$$

where $L_t$ is the true length and $L_s$ is the shortest Euclidean distance between the first and last point of the contour line (Figure 4). A value larger than 1 indicates a high degree of sinuosity, whereas values close to 1 indicate a low degree of sinuosity. Before calculating the sinuosity of the shoreline, a high pass filter was used to remove sinuous signals larger than 400 m. This was done to filter out larger-scale undulating patterns that are not enforced by the inner-bar rip channels (Castelle et al., 2015).

The metocean and topographic data collected at Cap Ferret or near Truc Vert are both located approximately in the centre of the study area. This data was assumed to be representative of wave and weather conditions along the entire study area. When constructing a BN (see next section), probabilities of a SZI occurrence must be compared to a probability of non-SZI. Therefore, a discretization in time is needed. A 1-hour time window was chosen to count the number of SZIs. To avoid a spurious distribution of non-SZIs, only daily hours between 7h and 21h were used.



**Figure 3.** (a) Example of a digital elevation model of Truc Vert beach on 29-Apr-2013, with the colorbar showing elevation amsl. The four cross-shore transects used to compute the inverse beach slope $IFS$ between 1 m and 3 m amsl are indicated by the dashed black lines. (b) Time series of $IFS$ and (c) summer shore-break related injuries.





**Figure 4.** (a) Example of a digital elevation model of Truc Vert beach on 29-Apr-2013, with the colorbar showing elevation amsl. The red line indicates the shortest Euclidean distance. Time series of (b) sinuosity $S$ and (c) number of summer rip-current related drownings.

## 3   Bayesian networks

### 3.1   Bayes' Theorem and BN structure

Bayesian networks (BN) are based on Bayes Theorem (Korb and Nicholson, 2010). This theorem states that probabilities of a certain event can be updated, given new evidence and can be stated as (Bayes, 1763):

$$P(h|e) = \frac{P(e|h)P(h)}{P(e)} \tag{3}$$




where $P(h|e)$ is the probability for a hypothesis $h$, given the evidence $e$. In Eq. (3), $P(e|h)$ resembles the likelihood and $P(h)$ corresponds to the prior probability of $h$ before any evidence was given. Dividing the numerator by $P(e)$ is a means of normalizing, so that conditional probabilities sum to 1. For example $P(h|e)$ could be the probability of a rip-current hazard, given that the tide was low.

A BN is a graphical representation of the probabilistic relations between a set of variables, using Bayes' Theorem to describe the relation between variables (Korb and Nicholson, 2010). The links between nodes represent the direct dependency between variables (or nodes). A constraint on linking variables is that links cannot return to the beginning node, completing the cycle. Therefore the graphical representation of a BN is often referred to as a directed acyclic graph. If there is an arc from variable A to B, variable A is termed the parent variable and variable B the child variable. The relation between variables is often

assumed to be causal, but is not necessarily the case. Once the structure is established, relations between variables are quantified according to conditional probability tables (CPTs), in the case of discrete variables. The probability of a value for a child variable is calculated for each possible value that the parent variable can take. Given that this is done for all variable nodes in the BN, two types of probabilistic reasoning become possible. Firstly, predictive reasoning, where a value is specified for each input variable. This results in a predicted probability for a target variable. Secondly, diagnostic reasoning is the other type for

which, for example, given a SZI the BN can specify the probability that it was low tide.

### 3.2    Construction of the rip-current and shore-break related BNs

Constructing a BN requires a trade-off between complexity and predictability. This is determined by the amount of variables chosen, the way variables are discretized and how the variables are linked. In general a simpler model is preferred over a complex model with the same performance, according to the principle of Ockham's razor (Jefferys and Berger, 1992). In this

section we describe the choices that were made related to this trade-off, using the BN software package Netica v. 6.05 (Norsys, 1998).

   Based on earlier work on the environmental controls on SZIs in southwest France (Castelle et al., 2019) and some preliminary BN tests, the rip-current BN is made of: i) a hazard component that depends on hydrodynamic forcing parameters $H_s, T_{02}, \theta, \eta$ and $d\eta$ and a morphological component $S$ and ii) an exposure component that depends on the hour of the day $H$, temperature $T$ and insolation $I$ (see Figure 5). The shore-break BN has a similar set-up, but the shoreline sinuosity is replaced by the inverse

foreshore slope ($IFS$), and the tidal gradient ($TG$) is replaced by tidal range ($TR$). Such structure was motivated by the fact that sometimes a simpler and computationally less expensive network can be obtained by adding so-called hidden or latent variables that limit the amount of links between variables or the amount of variables to include in the network (Russel and Norvig, 2010, p.817). In this case, we used two hidden exposure and hazard variables which are known to control life risk and

the amount of SZIs (Stokes et al., 2017).

   In order to compare the probability of an injury with the probability of a non-injury, injuries were counted per hour for all summers. Consequently, the variable injury count was discretized as a binary variable with two possibilities: no injury or an injury. Where the amount of injuries per hour exceeded 1, the cases were duplicated proportionally. Often hidden learned variables tend to be discretized with a small amount of bins, as they do not have any prior information available. After testing





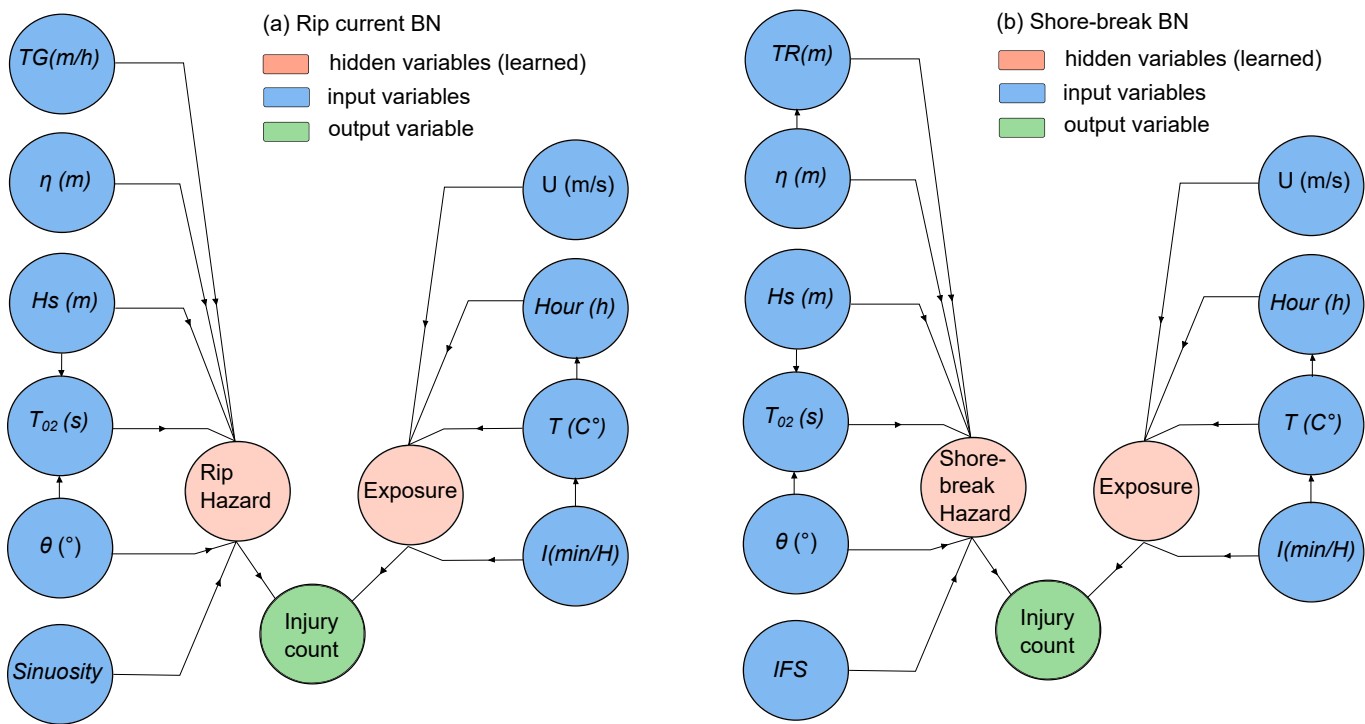

**Figure 5.** Bayesian Networks for (a) Rip current related injuries and (b) Shore-break injuries. Both BN are defined by input variables, hidden variables, an output variable and their linkages.

both BNs, 3 dummy bins were chosen for the exposure variable and 2 dummy bins for the hazard variable. The amount of bins chosen for the input variables determine the performance of the model to a larger extent. Therefore, different discretization options were tested, keeping an equal bin width. This is shown in Section 3.4.

### 3.3 Bayesian network training

The probability tables of the hidden variables were calculated by using an algorithm that calculates the most likely distribution of the data, given the probability distributions of the other variables. The expectation maximization (EM) algorithm is widely used in BNs to determine the most likely model given the data (Russel and Norvig, 2010). In a similar manner the algorithm finds the most likely value for occasionally missing data. The algorithm proceeds in the direction of the steepest gradient to find the minimum negative log likelihood for a model, given the data. The amount of bins used to discretize variables in the BN determines how well data is described. The larger the number of bins, the better it describes the data until the point where there might be a bin for each value. On the other hand, a larger number of bins degrades the prediction skill of the BN, as it becomes harder to predict the correct bin. A BN might be trained slightly differently from one run to another, because it is a probabilistic process. Therefore, we used $K$-fold cross-validation to eliminate any bias that single model runs might hold. After Fienen and Plant (2015); Gutierrez et al. (2015); Pearson et al. (2017) the cases were separated in $k$ random partitions,





where $n - n/k$ cases were used for training and calibration, and $n/k$ cases were left out for testing/validation. We used $k = 10$

so that test cases make up 10 % of the total data set. After 10 folds, mean values of performance metrics were taken to evaluate the performance of the BN.

### 3.4   BN performance metrics

Different performance metrics were used to address BN predictive performance. Here we used three relevant metrics: skill $sk$, log likelihood ratio $LLR$, area under ROC (receiver operating characteristic) $AUC$.

Skill was adapted from Fienen and Plant (2015) and was computed as:

$$sk = [1 - \frac{\sigma_e^2}{\sigma_o^2}] \times 100\% \qquad (4)$$

where $\sigma_e$ is the mean squared error between observations and the BN forecast and $\sigma_o$ is the variance of observations. The skill metric in Eq. (4) expresses how close predictions of an injury match with observations of an injury, with $sk = 1$ meaning perfect prediction.

Because skill is not an optimal measure for binary output variables, the $AUC$ (area under ROC curve) was chosen as a complementary metric (Marcot, 2012). $AUC$ is based on the ratio between the true positive predictions of the BN and the false positive predictions. Figure 6a shows the sensitivity on the y-axis (true positive rate) and the specificity on the x-axis (false positive rate) of a typical ROC curve from one of the model runs. If the dashed random classification line is equal to the ROC curve, this indicates that the model is not able to distinguish an injury from a non-injury. This corresponds to $AUC = 0.5$.

Figure 6b shows the confusion matrix on which the sensitivity and specificity is based.

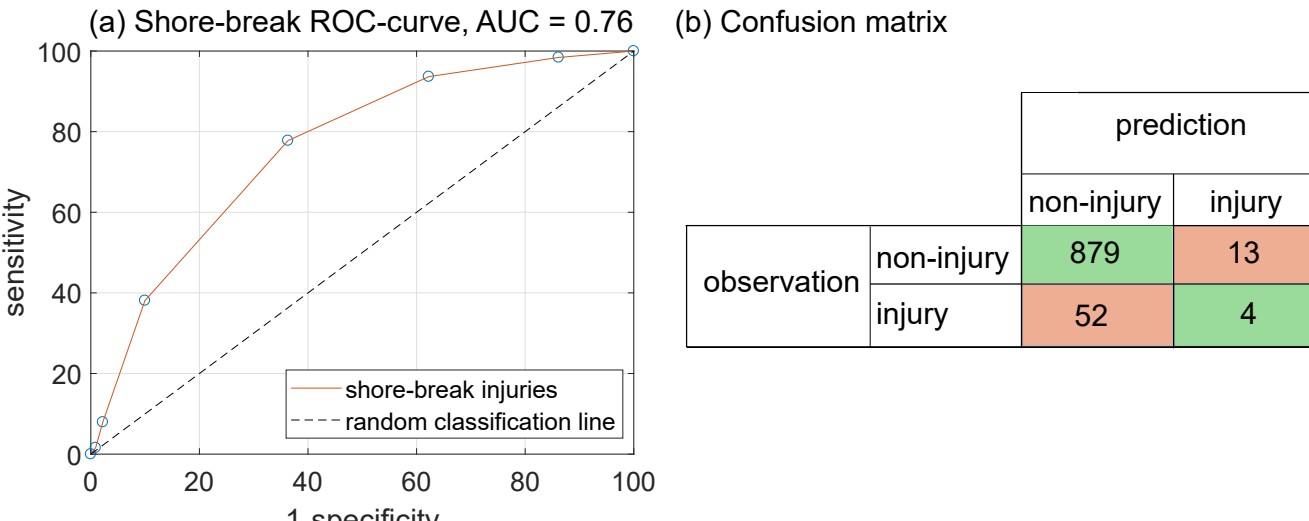

**Figure 6.** (a) Receiver operating characteristic curve (ROC) of the shore-break BN of one of the fold runs; (b) Confusion matrix with false positives in orange and true positives in green





The third metric is the log likelihood ratio ($LLR$), adapted from Plant and Holland (2011) and Fienen et al. (2013). The $LLR$ compares the prior probability of an injury with the posterior probability of a prediction, given the evidence (the input variables), which reads:

$$LLR = log_{10}(p(F_i|O_j)_{F_i=O_j'}) - log_{10}(p(F_i)_{F_i=O_j'}) \tag{5}$$

where $F_i$ is a forecast, in this case of a SZI, $O_j$ is an independent observation that was withheld from the forecast (e.g. a tidal elevation of -2.0 m). The $LLR$ expresses the change in likelihood due to certain evidence in the form of observations. A $LLR$ that exceeds zero indicates that the BN offers a better forecast than the prior probability. A $LLR$ that is lower than zero, indicates that the prior probability is a better forecast than the BN forecast. The $LLR$ can be calculated for each predicted case and each variable and can then be summed over the entire BN ($\sum LLR$). The $LLR$ penalizes wrong but confident predictions

more than wrong predictions that are uncertain (Plant and Holland, 2011; Pearson et al., 2017). Therefore, it is a suitable metric to verify whether the BN is over fitting or not.

Finally, in order to address how each input variable influences the target variable (SZI), the percentage of variance reduction $Vr$ that was caused by updating the BN based on the evidence was computed as:

$$Vr = \frac{V(F) - V(F|O)}{V(F)} \times 100\% \tag{6}$$

where $V(F)$ is the variance of a forecast prior to any evidence, and $V(F|O)$ is the variance of the forecast, given the new evidence. $V(F)$ and $V(F|O)$ are calculated as:

$$V(F) = \sum_{j=1}^{N} p(f_j)(f_j - E(f_j))^2 \tag{7}$$

$$V(F) = \sum_{i=1}^{N}\sum_{j=1}^{N} p(f_j|o_i)(f_j - E(f_j|o_i))^2 \tag{8}$$

where $p(f_j)$ is the prior probability of the $j$th forecast, $f_j$ is the current value of the $j$th forecast, $E(f_j)$ is the expected value predicted by the BN of the $j$th forecast, $p(f_j|o_i)$ is the predicted value of the $j$th forecast given the $i$th evidence case, $E(f_j|o_i)$ is the predicted value of the $j$th forecast given the $i$th evidence case, $M$ represents the number of evidence data and $N$ the number of predictions.

## 4 Results

### 4.1 BN performance


To find the best BNs, a varying number of bins was tested to evaluate the trade-off between calibration and validation. For calibration, the BN was used to make predictions of an injury based on the input variables of the training cases. For validation, predictions of an injury were made based on the input variables of the 10 % left-out cases. Generally an increase in complexity





leads to a decrease in predictive capability and vice versa (Fienen and Plant, 2015; Fienen et al., 2013). Figure 7 and 8 show

performance metrics for both the shore-break and rip-current BNs, respectively, as a function of the number of bins for the

input variables.

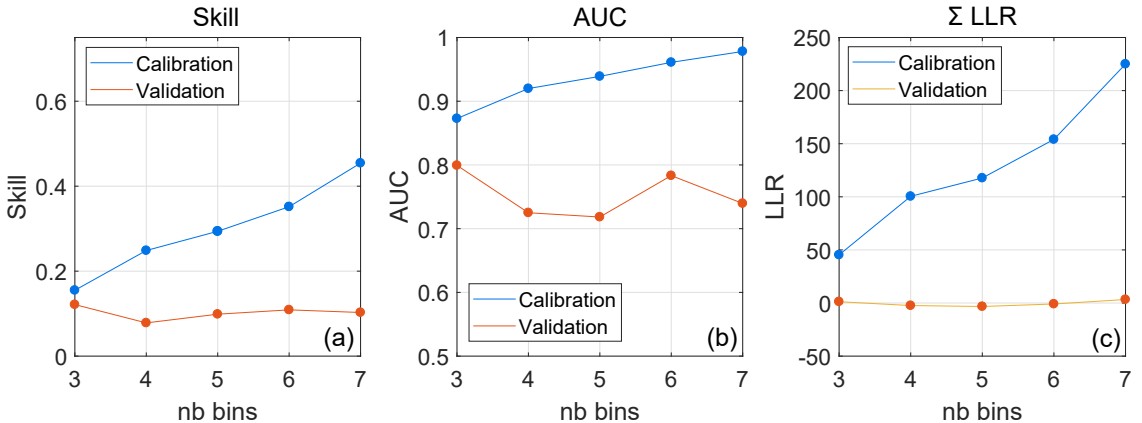

**Figure 7.** Performance metrics of shore-break BN as a function of the number of bins of the input variables and for validation and calibration : (a) skill $sk$, (b) area under ROC curve $AUC$ and (c) the summed log likelihood ratio $\sum LLR$.

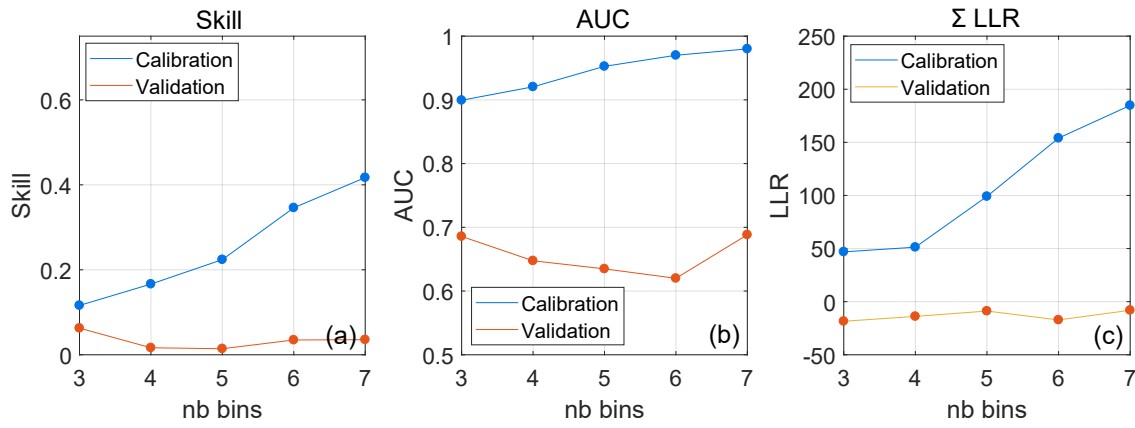

**Figure 8.** Performance metrics of rip-current BN as a function of the number of bins of the input variables and for validation and calibration : (a) skill $sk$, (b) area under ROC curve $AUC$ and (c) the summed log likelihood ratio $\sum LLR$.

It can be observed that the shore-break BN performs slightly better than the rip-current BN, as all performance metrics score better. Calibration results of the BNs are fair with $sk$ and $AUC$ ranging 0.15-0.43 and 0.89-0.98, respectively. Validation $sk$ is smaller and ranges 0.078-0.12 and 0.035-0.06 for the shore-break and rip-currents BNs, respectively. Validation $AUC$

are better, ranging 0.71-0.8 and 0.63-0.68 for the shore-break and rip-current BNs, respectively. The sum of the $LLR$ is systematically smaller than 0 for the validation of both BNs. This is either an indication that the prior estimate is on average




better than the prediction of the model, or that there are anomalous cases where the wrong but confident prediction is heavily punished by highly negative $LLR$ values that result in a negative or near zero $LLR$ sum.

Table 4 and 5 show that, depending on the number of bins chosen, model predictions are better than the prior probability
estimate 62.21% - 79.9% of the time. When 5 bins are chosen for the rip current BN, 79% of the time the model prediction is of added value. When 5 bins are chosen for the shore-break BN, 72% of time the model prediction performs better. This shows that the negative and near zero sums of the $LLR$ displayed in Figure 7 and 8 must be caused by anomalous events (the remaining percentages) that are confidently predicted wrong.

**Table 4.** The percentage of $LLR > 0$ for rip current SZIs, indicating whether the prediction of the model is better than the prior probability

| Nb bins | Prediction rip current SZIs (% LLR>0) |
|---------|---------------------------------------|
| 3 bins  | 73.86% |
| 4 bins  | 77.30% |
| 5 bins  | 79.90% |
| 6 bins  | 61.60% |
| 7 bins  | 62.21% |

**Table 5.** The percentage of $LLR > 0$ for shore-break SZIs, indicating whether the prediction of the model is better than the prior probability

| Nb bins | Prediction shore-break SZIs (% LLR>0) |
|---------|---------------------------------------|
| 3 bins  | 70.18% |
| 4 bins  | 70.08% |
| 5 bins  | 72.56% |
| 6 bins  | 65.10% |
| 7 bins  | 69.50% |

The amount of bins was varied from 3 to 7 bins to choose the best trade-off between complexity and accuracy. Only the
number of input variable bins were adjusted, keeping the output variables exposure, hazard and injury count the same. In general, an increase of the number of bins leads to a better descriptive capability and a worse predictive capability (Fienen and Plant, 2015). When the number of bins is increased from 3 to 4, a small decrease in $sk$ and $AUC$ can be noticed. However, a further increase in the number of bins does not significantly lead to worse $sk$, $AUC$ or $\sum LLR$. $AUC$ and $sk$ show a small increase at 6 bins for the shore-break BN (Figure 7a,b) and at 6 / 7 bins for the rip BN (Figure 7). Contrary to what is generally
observed, validation $sk$, $AUC$ and $\sum LLR$ did not drop dramatically when complexity was increased. However, the percentage $LLR > 0$ did drop for both BNs, when increasing the number of bins, which is in line with expectations.

### 4.1.1 Input variable sensitivity

Figure 9a shows the sensitivity of the shore-break BN to the input variables (6 bins) and to the hidden variables exposure and shore-break hazard. For the shore-break BN, the learned variable exposure ($Vr$ =23.5%) has the strongest influence followed
by the shore-break hazard variable ($Vr = 10.9\%$). This suggests that exposure of water users has a more dominant control on the injury count than the shore-break hazard forcing variables. This is also reflected in Figure 9b, where the hour of the day, temperature and insolation have larger values for $Vr$. These variables are followed by tide elevation which is the most important shore-break hazard control ($Vr = 0.17\%$). Consequently, mean wave period ($Vr = 0.07\%$), tidal range ($Vr = 0.06\%$), significant wave height ($Vr = 0.05\%$) and wind speed ($Vr = 0.039\%$) follow. Even though beach profiles were taken at only


one location along the coast, the inverse foreshore slope $IFS$ still has a noticeable impact ($Vr = 0.065\%$) on the shore-break hazard. Wave direction is least sensitive to the injury count with $Vr = 0.025\%$.

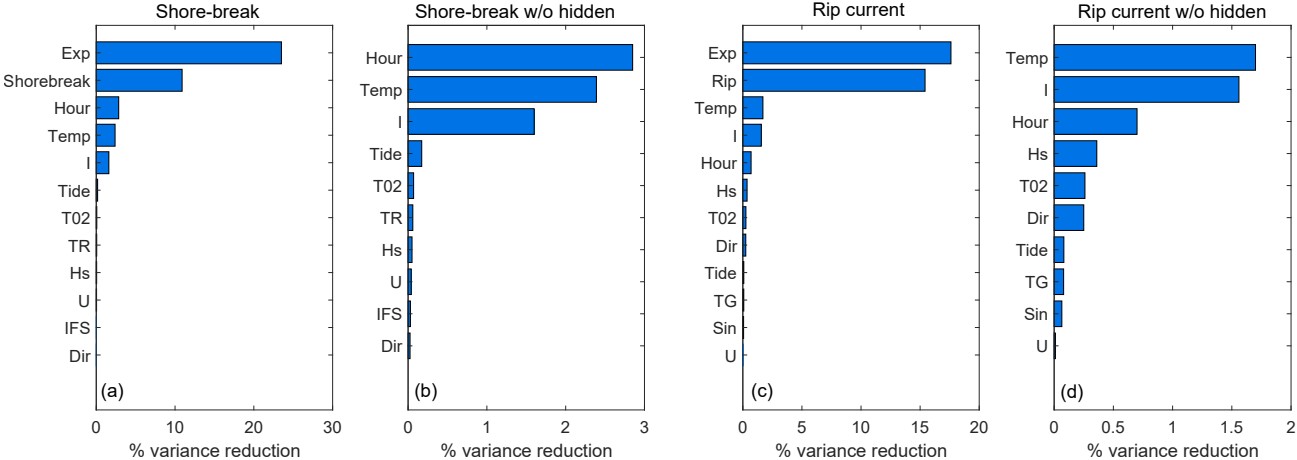

**Figure 9.** Variance reduction $Vr$ without hidden learned variables for the shore-break BN (7 bins) (a) with hidden learned variables and (b) without learned hidden variables and for the rip-current BN (6 bins) (c) with hidden learned variables and (d) without hidden learned variables

Figure 9c shows the sensitivity of the rip-current BN to the input variable (7 bins) including the hidden variables exposure and rip-current hazard. Rip-current hazard ($Vr = 17.6\%$) and exposure ($Vr = 15.4\%$) have similar influence, even though parents of exposure (insolation, temperature and hour) are more dominant in Figure 9d. It suggests that it is the combined effects of input variables that cause a rip current (e.g. tide, wave direction and wave height) that have a strong influence on the occurrence of drowning incidents. This is different from what was observed for the shore-break BN. Most sensitive input variables for the rip-current BN are insolation ($Vr = 1.67\%$), temperature($Vr = 1.7\%$) and hour of the day ($Vr = 1.56\%$). These are followed respectively by the hazard-related variables, tidal elevation, wave direction, significant wave height, tidal range, wave period and sinuosity of the shoreline. $H_s$ and $T_{02}$ have the highest $Vr$ with 0.36% and 0.26%, suggesting that wave energy is the most important control on rip-current hazard. They are closely followed by wave direction with 0.25%. Shoreline sinuosity $S$ reduces variance by 0.065%.

### 4.2 Scenario analysis

Apart from the predictive ability of a BN, probabilistic scenario analysis can be a useful tool to understand how multiple variables interact. Figure 10a shows the prior joint probability distribution of the shore-break BN without updating based on any evidence. The two hidden variables, exposure (3 bins) and hazard (2 bins) do not contain any prior information and thus have equal probabilities for each bin. Figure 10b shows the joint probability distribution for a trained shore-break BN updated for the evidence that there was a shore-break SZI. In the latter, the distribution of tidal elevation shifts towards high tide.


Additionally, there is a shift towards higher mean wave periods ($T_{02}$) and a slight increase in probabilities of larger wave heights ($H_s$). Furthermore, temperature, hour of the day and insolation show a pronounced shift towards higher temperatures,
less cloud cover and the afternoon between 14h and 16h30.

Noteworthy, when the BN was updated for larger wave heights, the probability of a shore-break related injury increased. However, when the BN was updated with the evidence of an injury, intermediate wave height bins 0.75-1.5 m and 1.5-2.25 m showed increased probability. This supports our hypothesis that large shore-break waves ($H_s > 2.5$ m) can discourage bathers from entering the water, which results in less SZIs despite that the shore-break hazard is increased.

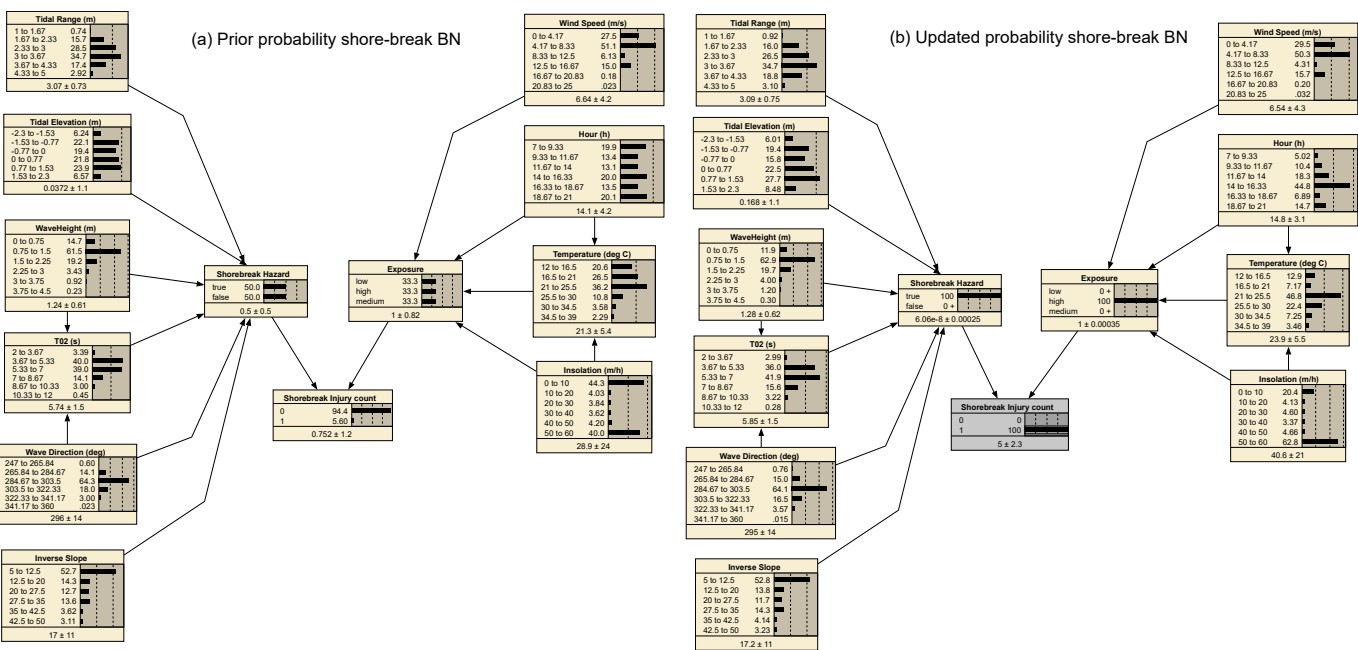

**Figure 10.** (a) Prior probability distribution of the shore-break BN; (b) Updated probability distribution where the probability of an injury occurring was set to 100%

Apart from an injury hindcast scenario, various other scenarios provided insight in variable interaction. For the shore-break BN an intermediate slope during a shore-break hazard is slightly more likely during low tide (18.9%) than during high tide (13%) (see Figure 11a and b). This might be explained by the hypothesis that different parts of the foreshore slope are active during different tidal elevation levels.

A similar scenario analysis was performed for the rip current BN. Figure 12a shows prior probabilities of the rip-current BN.
The rip current BN shows that according to the prior probability a shore-break injury (5.60%) is more likely than a rip-current related injury (3.23%). Figure 12b shows the updated probability distributions for the rip current BN, given that there is a 100% chance of an injury. Larger tidal gradients (both negative and positive) show a slight increase in probability, supporting the hypothesis that a rapid change in tidal elevation can surprise water users by driving the rapid onset of rip current activity. Low tides become slightly more likely when there is evidence of a rip-current related drowning incident. Increased probabilities


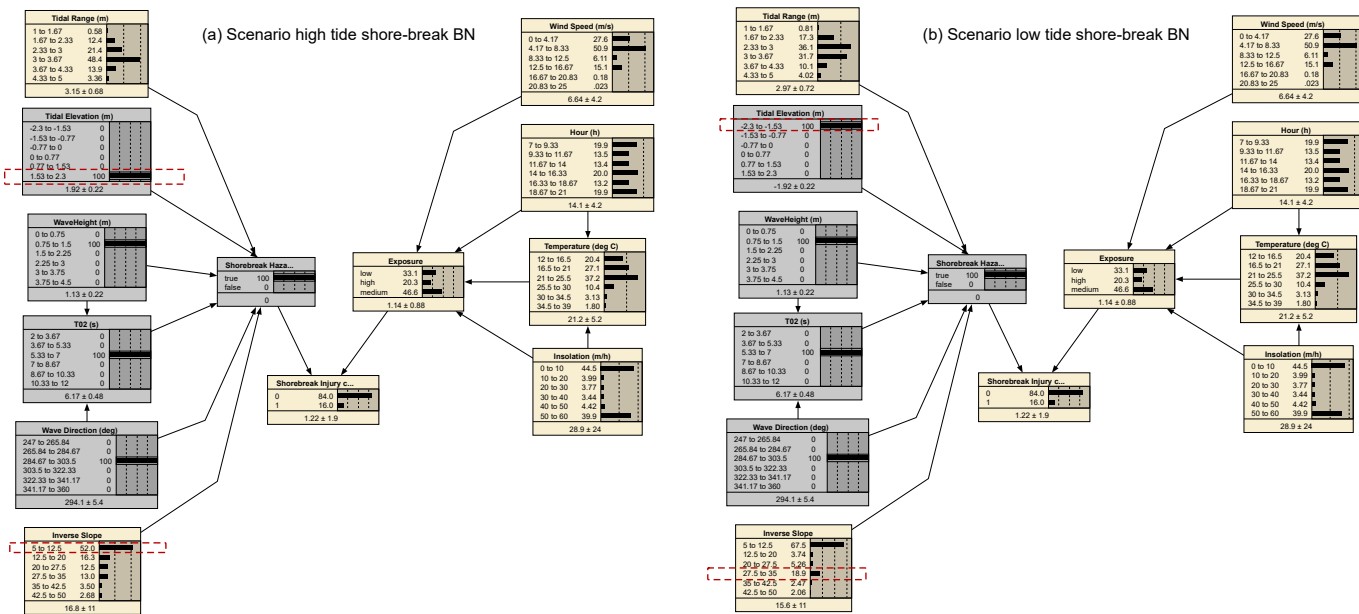

**Figure 11.** (a) Scenario with high tide resulting in a steep slope with a high probability; (b) Scenario with low tide resulting in a steep slope with a high probability and a intermediate slope with a slightly elevated probability

of larger $H_s$ and $T_{02}$ suggest such drowning incidents occur for increased incident wave energy. Wave direction shows small increase in drowning probability for the NW oriented directions. More sinuous shorelines (larger $S$ values) show increased probability of rip-current related drowning, indicating that more alongshore-variable surf zone morphology increases the rip-current hazard. Wind speed has only little influence, although low wind speeds are slightly more likely during a drowning incident. Furthermore, the hour of the day shows a distribution that corresponds with the expected beach attendance. However,

there is a disproportionate peak in the evening between 19h and 21h. Furthermore, the highest peak is much earlier between 13h-15h compared to shore-break injuries. This might be explained by the fact that in summer more high tides with larger tidal ranges occur during late afternoon (Castelle et al., 2019). Temperature and insolation show comparable patterns to the shore-break BN with warm sunny days between 20-28 C° having the highest probability.

        Other rip-current BN scenarios were tested, providing insight in variable interactions. For instance, an interaction between

the magnitude of shoreline sinuosity $S$ and wave angle of incidence was explored, given average wave energy conditions and a 100% chance that there was a rip hazard (Figure 13a and b). It can be observed that a low sinuosity (1-1.06) is correlated with higher probabilities for the shore-normal angle of wave incidence (around 279°) and a high sinuosity is associated with a larger probability for more NW angles of incidence (295.43-311.57°).



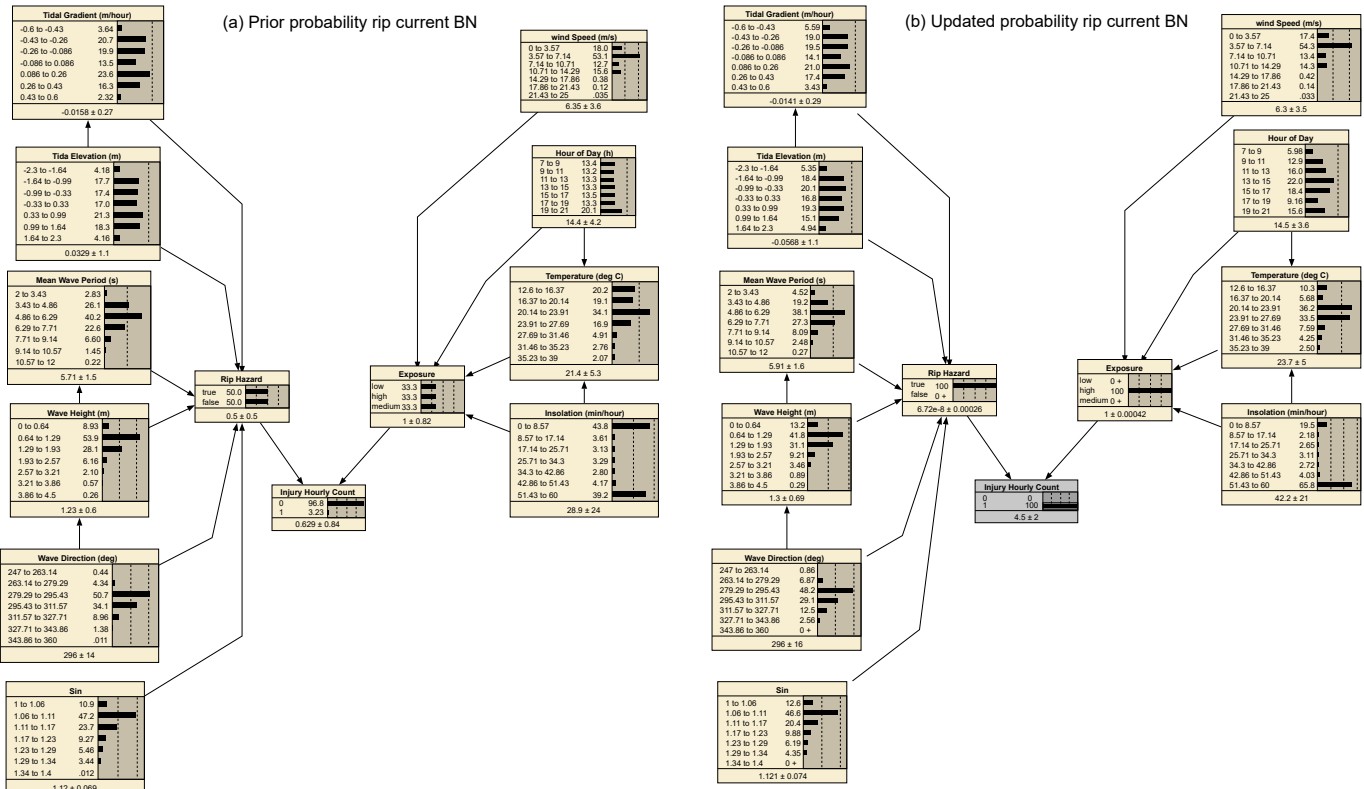

**Figure 12.** (a) Prior probability distribution of the rip-current BN; (b) updated probability distribution of the rip current BN when the probability of an injury occurring was set to 100%

## 5   Discussion

### 5.1   BNs as a predictive tool for SZIs

Contrary to previous works, a same BN approach was used to address shore-break and rip-current related SZIs co-existing at a given site. Two separate BNs were created for shore-break SZIs and rip current SZIs. This allowed to use different beach morphology metrics based on prior understanding of the physics of shore-break waves and rip current dynamics. In addition, in line with Stokes et al. (2017) two hidden variables (exposure and hazard) were introduced for both BNs to decrease the amount of connections and increase BN efficiency. Importantly, Doelp et al. (2019) used population data to test a SZI ratio, normalized by the population, in addition to the binary injury likelihood. Although results were not dramatically improved in Doelp et al. (2019), including accurate water user data should improve BN model predictions along the Gironde Coast. However, such data does not exist and will require future research effort.

Performance metrics indicate that the BNs improve prior estimates, but that BNs still have a significant percentage of wrong but confident predictions. This is due to over fitting, which is a common issue with training a BN on rare events (unbalanced




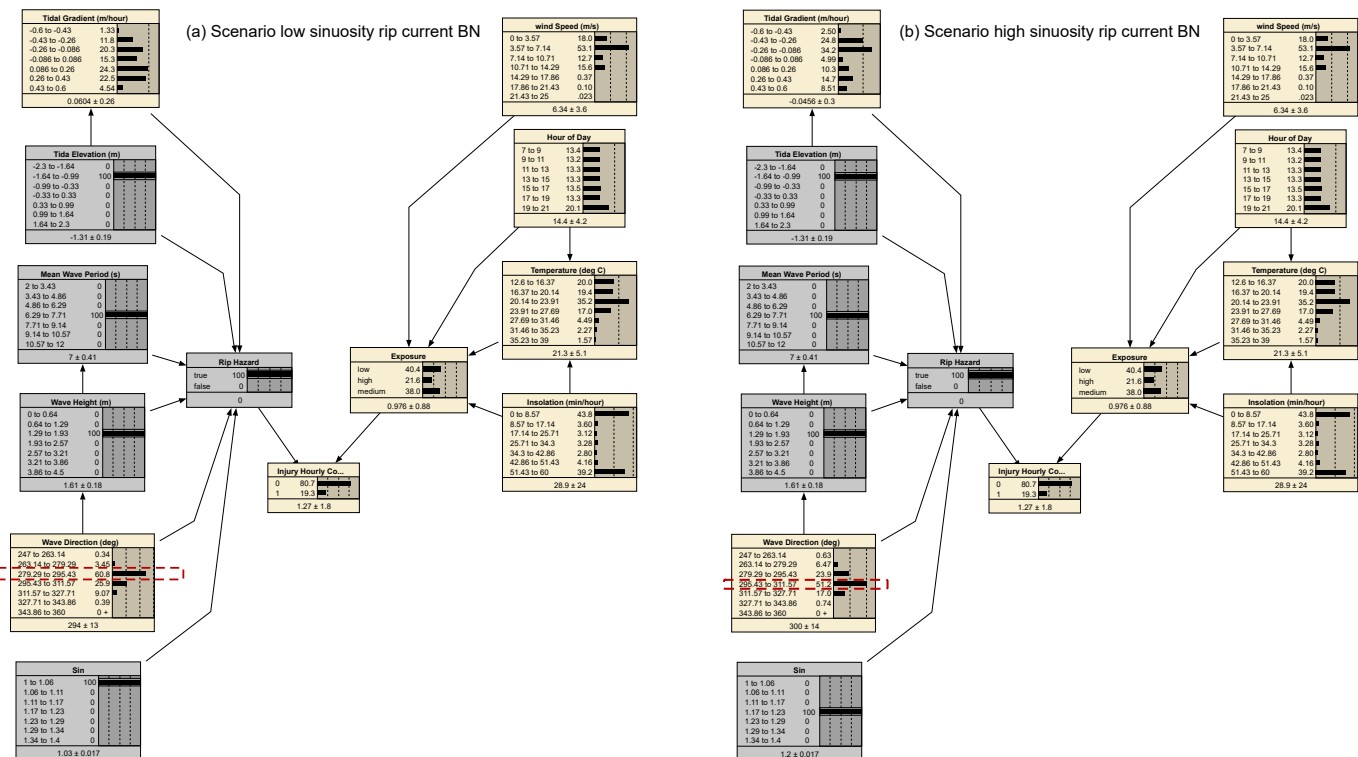

**Figure 13.** (a) Scenario with low sinuosity and shore-normal (around 279°) wave direction; (b) Scenario with medium sinuosity and NW wave angle

dataset) (Cheon et al., 2009). When the primary objective of a BN is prediction rather than description, a synthetic data set can be created with an even distribution of events although it degrades the BN descriptive ability. A similar suggestion to cope with this problem, is to remove anomalous confident but wrong predictions (Doelp et al., 2019). Another issue limiting the BN predictive ability is that simple beach morphology parameters $S$ and $IFS$ were derived from a single site (Truc Vert).

Summer beach profiles acquired along the different beaches should help improving $IFS$ estimation and, in turn, prediction of rip-current drowning incidents. Similarly, optical satellite images should be explored to derive beach sinuosity $S$ at the different beaches along the coast. As indicated earlier, an estimation of beach attendance or water users should also improve BN predictive ability. Therefore, at this stage other tools should be used for life-risk prediction, like for instance models based on simple correlations between meteorological, oceanographic conditions and the incidence (Lushine, 1991; Lascody, 1998;

Dusek and Seim, 2013) or on the numerical modelling of rip flow speed (Austin et al., 2013). Recently, using the same SZI dataset a logistic regression model was found to predict the risk of drowning the Gironde up to three days in advance with good skill (Tellier et al., in revision).

Lastly, there were 1565 unknown injuries that could not be retrieved to either a shore-break or a rip-current related injury. Theoretically, a well trained BN could estimate which of the SZI is more likely based on the environmental conditions. How-





ever, since the BNs are still limited in prediction this should be explored in the future using improved BNs. Such BN could help to retrospectively improve SZI statistics along surf coasts.

## 5.2 Environmental controls on SZIs and implications for beach safety management

In other studies frequency analysis was used to identify disproportionate environmental conditions during SZIs (Scott et al., 2014; Castelle et al., 2019). Some of the BN results are essentially in line with previous work. In short, more SZIs are observed
for warm sunny days with light winds. Rip-current related drowning incidents increase with increasing incident wave energy (height and period), more shore-normal incidence, and lower tide level. In contrast, shore-break related injuries are sustained at high tide levels and moderate wave height. In addition to previous work, here we proposed a method to quantify the role of beach morphology on SZIs. Beach sinuosity $S$, which is a measure of the alongshore variability of surf zone morphology, and inverse beach slope $IFS$ were found to influence the occurrence of rip-current related drowning incidents and shore-
break-related injuries, respectively. These results are in agreement with current knowledge of rip flow intensity increasing with increasingly alongshore-variable surf zone morphology (Moulton et al., 2017), and shore-break waves occurring for steeper beach face (Battjes, 1974; Balsillie, 1985). We also found that rapid, positive or negative, change in tide level elevation (large $d\eta$) increase the probability of rip current related drownings. This can explain why in previous work large tidal ranges were found to result in more rip current drownings (Scott et al., 2014; Castelle et al., 2019). Although this essentially applies to
meso- to macro-tidal beaches and given that $d\eta$ does not affect the hazard posed, this demonstrates that rapid changes in tidal elevation driving the rapid onset of rip current activity can surprise unsuspecting bathers and carry them offshore. To our knowledge, the rapid onset of hazardous surf zone currents for days with large tidal range is not emphasized in current public safety awareness campaigns on rip currents.

In addition to the primary environmental controls on SZIs, in this study it was for the first time possible to identify the
interaction between multiple input variables. For instance, it was found that it is the combined effects of tide elevation, wave direction and wave height that control rip-current hazard. In other words, even if you have shore-normally incident waves near low tide, if wave height is very small, there is no hazard and consequently a low probability of rip-current related drownings. Such interactions, which were not possible to address in previous work (Scott et al., 2014; Castelle et al., 2019), are in line with the understanding of rip flow response to wave and tide conditions (Castelle et al., 2016). Similarly, evidence was found
that rips can be activated during various angles of wave incidence, depending on the degree of shoreline sinuosity $S$. This is also in line with observations and model outputs showing that, for the same obliquely incident wave conditions, rip cell circulation are transformed into an undulating, less hazardous, longshore current for weakly (small $S$) alongshore variable surf zone morphology, while rip cell circulation can be sustained for deep rip channels (MacMahan et al., 2008; Dalrymple et al., 2011). This shows that BNs including a wisely pre-defined hidden hazard variable can provide insight into the influence of the
primary input variables and their interactions on the hazard posed. Therefore, it could also be applied to other injuries, e.g. related to surfing activity, for which the causes (e.g. environmental, behavioural) and their interplay are poorly understood.

Studies addressing the environmental controls on shore-break related SZIs are scarce (Puleo et al., 2016; Doelp et al., 2019) compared to drowning studies. The shore-break BN developed herein for the Gironde coast suggests that, with decreased





exposure for $H_s > 2.5$ m, large surf, and thus heavy shore-break waves at the shoreline, discourage the beachgoers to enter
the water near high tide. Importantly, this was not observed for rip-current related drownings, which have a tendency to occur
at low tide with the inner surf zone located on a much more gently sloping part of the beach profile. We hypothesize that in
such less adverse conditions, beachgoers are less discouraged to enter the water, as opposed to facing large shore-break waves.
However, further investigation on beachgoer behaviour in the presence of shore-break waves is required to test this hypothesis.

In addition, our variable sensitivity analysis indicates the shore-break related injuries are more controlled by the exposure
than by hazard, contrary to rip-current related drowning for which life risk is approximately equally distributed between hazard
and exposure. This indicates that shore-break injuries are more likely to occur during busy days, whether moderate or heavy
shore-break conditions are present. In contrast, the presence of intense rip currents is critical to drowning incidents.

## 6 Conclusions

A Bayesian network (BN) approach was used to model life risk and the controls and interactions of environmental (metocean
and morphological) data on SZIs along a high-energy meso-macrotidal coast where shore-break and rip-current hazards co-
exist. In line with previous work, the BNs show limited predictive skill. Although the shore-break and rip-current BNs improves
prior estimates, they still have a large percentage of wrong but confident predictions, which is not tenable for life-risk prediction
on beaches. However, the BNs provide fresh insight into the different environmental controls, their interactions, and their
respective contribution to hazard and exposure. For the first time, the respective contributions of exposure and hazards to
the overall life risk were quantified, showing the shore-break related injuries are more controlled by the exposure than by
hazard, contrary to rip-current related drowning for which contributions are approximately equal. These results can guide the
future development, or modification, of public education messaging, particularly on the shore-break hazard that received little
attention so far compared to rip currents, despite the large number of severe injuries sustained in shore-break waves along the
Gironde Coast. We advocate that such BNs should be developed in parallel with other risk predictors showing high predictive
skill but providing much less diagnostic Tellier et al. (in revision).

*Data availability.* The wave buoy data are publicly available through the French Candhis network operated by CEREMA. Weather station
and tide gauge data are available from the Météo France Radome network and the SHOM, respectively. Wave hindcast is available from
the MARC platform (Modelling and Analysis for Research in Coastal environment) at https://marc.ifremer.fr.The injury data collected on
beaches are not publicly available due to restriction from the French National Committee for the Protection of Data Privacy.

*Author contributions.* BC and ET designed the research presented here. EdK designed the experiments, performed the modelling experiments
and analysed all the results. BC and ET consulted on the experiments. EdK prepared the manuscript, and BC and ET reviewed and edited the
manuscript. Funding provided by BC.





*Competing interests.* The authors declare that they have no conflict of interest.

*Acknowledgements.* We thank Dr. Bruno Simonnet who contributed to the injury data collection. Constructive suggestions to improve read-
ability by Rieke Santjer were much appreciated. This study includes the beach monitoring study site of Truc Vert labelled by the Service
National d'Observation (SNO) Dynalit, which surveys are financially supported by Observatoire de la Côte Aquitaine (OCA) and SNO
Dynalit.



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
