# Peer review of "A Bayesian network approach to modelling rip current drownings and shore-break wave injuries"

_Natural Hazards and Earth System Sciences, 2021_

## Referee Comment (RC2)

Review of **A Bayesian network approach to modelling rip current drownings and shore-break wave injuries**
NHESS-2021-36

February 2021

Authors analyse two beach safety-related hazards in the Atlantic French coast and propose a BN-based model to describe/predict them as a function of a hazard and an exposure component, each one comprising relevant variables characterising environmental (hydrodynamic and climatic/weather) conditions. The topic is in line with one of the targets of NHESS and, in this sense, the manuscript can be of interest for many NHESS readers.

In what follows, some observations/comments/suggestions are given.

**General comments**

**[1]** The proposed model includes different environmental variables to characterize the hazard and exposure components. With respect to the ***exposure component***, the implicit hypothesis is that the meteorological conditions together the time of occurrence should indicate the beachgoers affluence. Although it seems a reasonable assumption, it would be recommendable to perform an independent validation because the errors/uncertainty in this "submodel" (weather conditions controlling number of beachgoers) should affect the predictability skill of the overall model (dependence of the probability of SZI on number of beachgoers). In a previous paper (Castelle et al 2019), the authors concluded the need of better quantify exposure to better explain SZI. Is there any curve of affluence for the studied beaches which can be used to "validate" the conceptual model of exposure?

**[2] Line 137.** Please define "tidal gradient".

**[3] Line 148.** Please specify the number of the "Figure".

**[4]** Figures 3 and 4 can easily be combined in just one.

**[5] Lines 196-198.** One of the implications of discretizing injuries in binary form (and duplicating cases when exceeding 1) is to artificially augment data to be used in the BN. May this affect the real representativeness of environmental conditions affecting SZI? Also, this may also artificially increase the BN predictive performance.

**[6] Figure 6.b.** The confusion matrix indicates a poor performance of the model to predict injuries. I assume that this matrix corresponds to one of the tested fold runs (in fact, one with the poorest performances). However, as it is presented it seems that this is the overall performance of the model, which if so, it indicates the absence of reasonable predictive model. Please put the matrix in the right context.

**[7]** You propose three different metrics to measure the BN predictive performance. Do we really need to use all of them to identify the best number of bins to be used in the model? If so, how their results must be jointly interpreted (integrated)?

**[8] Line 252**. M is defined, but where do you use it?

**[9] Line 278-279**. So, finally, how many bins are selected for each BN. Looking to the final BNs (figures 10 & 12), you have selected 6 for shore-break and 7 for rip-current. However, the variation in prediction for both when changing from 6 to 7 is almost the same. Why didn´t you use the same number of bins in both BNs? This should give more consistency to the analysis since input variables which are almost the same will be discretized in the same way.

**[10] Line 258**. Are you using "complexity" to refer to the number of bins used. I would not say this is complexity but "level of definition" (or similar). I understand "complexity" by the number of components/variables used in the model.

**[11] Line 289**. You mention that beach profiles were taken only at one location. Unless that the beach is alongshore uniform, this may significantly affect to the role of IFS within the BN. Since you are using sinuosity (to predict rip SZI) as a measure of beach departure from alongshore uniformity, I presume that the slope will change along the beach. Do you have an estimation of the range of variation of IFS? Maybe this is the reason for the very low contribution of IFS to the variance.

**[12] Line 293**. I think that figures referring % variance of Exposure and Hazard are wrong (figure 9c). Exposure is the larger than hazard.

**[13] Figure 9**. Can you use the same scale in the X-axis for both hazards for an easier visual comparison? Are figures 9b and 9d just a zoom of figures 9a & 9c after removing the first two contributions (hidden variables)? If yes, please indicate, as it is written it suggests they were obtained independently.

**[14] Lines 294-301**. When you compare the contribution of hazard latent variables to the variance for shore-break and rip-current hazards, you obtain a larger contribution in the case of rip than for shore-break. May this indicate that the built model (selected variables) for the shore-break is worse than the one for rips? (e.g. the above-mentioned potential effect of neglecting variations in IFS -comment [10]-)
**Line 297.** If we compare the %variance of exposure variables (hour, temp, I) in both hazards (fig 9b & 9d), they are very similar (different ordering but same order of magnitude).
**Line 300.** Formally, you are not including wave energy as a variable in your BN. Consider that wave energy will involve a non-linear combination of H and T and the observed contribution of these individual variables may significantly vary when combined to characterize wave energy.

**[15] Line 311**. Where the update for larger wave heights (and the resulting change in % of shore-break injuries) can be seen?

**[16] Lines 315-318**. I would say here that the most likely IFS during a shore-break hazard is a steep slope whatever the tidal elevation is. Then, if you want you can highlight secondary differences in IFS, but in any case you are just concentrating in one single class for intermediate IFS (27.5 to 35), but when you refer to intermediate you could also include (20 to 27.5) and then, the probability at both tide levels would be almost the same. Furthermore, trying to draw any conclusions about IFS at low tide does not seem to make much sense, since IFS is measured above mean sea level.

**[17] Lines 321-323.** What are larger tidal gradients? If we consider the three central bins as representative of medium-low gradients, they are concentrating a similar % of occurrence. In fact, if we compare the probability distribution is almost similar to the prior one (Fig 11a).

**[18] Lines 326-328.** It is not clear from Figure 12b that more sinuous shorelines show increased probability of injuries. The updated distribution of Sin is quite similar to the prior distribution. Are the small changes detected large enough to support your conclusion?

**[19] Lines 329-332.** I disagree that your analysis is really reflecting that the peak of rip injuries (13h – 15 h) is much earlier than the one for shore-break one (14 to 16.33). Both bins overlap, which may be associated with the comparison of different bins resulting from using 6 classes in shore-break and 7 classes in rip for a same variable (Time).

According to Castelle et al (2019) "*For low TR, daily minimum tide elevation, which is when channel rip activity is maximised, tends to occur during the patrolled hours in the mid-to-late afternoon (Fig. 12b) when beach attendance (exposure) is maximised*". This does not seem to fully support your conclusion.

**[20] Legend of Figure 13.** Please change the legend to something similar to the one used in Fig 11 (e.g. Scenario with low sinuosity resulting in a higher probability of shore-normal wave direction). As it stands, it seems that you build the scenario by fixing booth sinuosity and direction.

**[21] Lines 347-348.** I agree with this comment. It would be interesting to assess the profile of people injured by shore-break and rips to identify potential factors affecting their relative exposure.

**[22] Lines 359-360.** See also the combination of video images and numerical modelling to help managing beach safety (Jiménez et al. 2007. Beach recreation planning using video-derived coastal state indicators. Coastal Engineering, 54, 507-521).

**[23] Section 5.2.** Please adjust comments on the role of different environmental factors according to your response to previous comments [e.g. 16, 17, 18, 19]

---

## Author Response (AR1)

**REVIEWER #1**

**Reviewer comments**

Our response

*Changes in the revised manuscript*
* * *
**General comments**

**This paper addresses the occurrence of rip current drownings and shore-break related injuries, and seeks to develop predictive and diagnostic models in the form of Bayesian Networks to model the occurrence of such incidents under different environmental conditions. The paper addresses a scientific question of relevance to NHESS, and uses a relatively novel way to model the occurrence of bathing incidents. In general, the paper is well written and structured, and the analysis of the data is well executed. The presented models seem to provide some skill in predicting the occurrence of the incidents, however, the skill varies considerably depending on which skill metric they present. Of most concern, they present a confusion matrix (Fig. 6b) which indicates that the model provides a very high rate of false negatives and only correctly predicts 4 out of the 56 observed injuries. If this table is correct, then the models are worse than useless as the vast majority of the time that injuries occur would be predicted to be safe. Please check this table and if the results are correct, the models need re-designing to a point where very few false negatives occur, otherwise the poor performance of the model undermines all the other findings.**

We thank the reviewer for their support for publication and for their insightful comments. Below you will see that all the comments have been carefully considered, and that the major issue on the confusion matrix, which was misleadingly included in the manuscript, has been address.

**Specific comments**

- **Line 6: "hidden hazard and exposure variables". It is not clear to the reader what is meant by 'hidden variables' at this stage, so I would suggest elaborating on this in the abstract if it is important to the methodology, or not mention 'hidden' in the abstract at all.**

We agree this part of the abstract now reads L6-8: *Each BN included two so-called 'hidden' exposure and hazard variables, which are not observed yet interact with several of the observed (environmental) variables, that in turn limit the amount of BN edges*

- **Lines 7-8: "Validation (prediction) results slightly improve predictions of SZIs with a poor to fair skill based on a combination of different metrics." It is not clear what is meant by this sentence; validation is used to prove the goodness of fit of a prediction, while calibration is used to improve the goodness of fit, so it is unclear how the validation results could have improved the model performance. Perhaps you mean some form of**

**calibration improved the performance? Please re-word this sentence to make its meaning clearer.**

This sentence was rephrased into L9 : *Results show a poor to fair predictive ability of the models according to the different metrics.*

- **Lines 9-10: "Shore-break related injuries appear more predictable than rip current drowning incidents as the shore-break BN systematically performed better than the rip current BN." This logic is not quite right – what this result shows is that your model did not perform as well for rips as it does for shore-break injuries, but unfortunately that does not mean that rips are necessarily less predictable (although that may be true). i.e. a different type of model may find rips easier to predict than shore-break injuries. I suggest changing the sentence to something like: "Shore-break related injuries appear more predictable than rip current drowning incidents using the selected predictors within a BN, as the shore-break BN systematically performed better than the rip current BN"**

Thank you for this suggestion. This sentence was replaced accordingly in our revised manuscript.

- **Lines 18-20: "Rapid change in tide elevation during days with large tidal range was also found to result in more drowning incidents, presumably because it favors the rapid onset of rip current activity and can therefore surprise unsuspecting bathers." Is it not also possible that stronger tidal currents (not rip currents) could play a role here? This is certainly observed in parts of the UK where beaches are found near to estuaries/inlets or where tidal flows are very strong. In most cases, tidal currents are very weak compared to rip current flows however, in which case please elaborate on why you discount tidal currents so easily.**

The reviewer is right. According to a similar comment made by Reviewer #2, the presence of tidal currents at some beaches is now discussed. However, given that it is only hypothesis we discuss this in the core of the paper but do not expand on this in the abstract, and removed reference to beachgoers being potentially surprised by rapid changes in water level. This also allowed us to shorten the abstract.

- **Line 36: "generally lower tide levels" I would suggest including a citation to Scott et al (2014) 'Controls on macrotidal rip current circulation and hazard', which addressed this topic in some detail**

Done.

- **Line 50: "A related challenge based on current research is filtering the effect of how water users choices are influenced by environmental conditions (e.g. wave height Hs)." I would suggest also looking at Stokes et al (2017) 'Application of multiple linear regression and Bayesian belief network approaches to model life risk to beach users in the UK' who observed certain parameters were related to higher water user exposure (even beach morphology), and in some cases also higher hazard**

We agree, we added L55-56: *For instance Stokes et al. (2007) found that beach morphology type has*

*an impact on the number of water users.*

- **Line 53: "Finally, the respective contributions of hazard and exposure to the overall life risk for shore-break waves and rip current are virtually unknown." I would suggest explaining the three components (exposure, hazard, and life risk) briefly early on in the introduction, and how they relate to one another. For instance, see explanation in Stokes et al (2017). This is not widely understood I think, and therefore warrants brief explanation.**

We agree. This concept is now explained in the first paragraph of the introduction section L24-29: *Wave-dominated beaches offer a playground for a variety of activities, but at the same time they pose a threat to water users. Following (Stokes et al., 2017) a conceptual definition of life risk at beaches can simplify in terms of the number of people exposed to life threatening hazards. As a result, a beach with a relatively high hazard level could exhibit a low level of risk if the number of beach users is low and vice versa. This way, the level of life risk can be modelled indirectly by estimating hazard and exposure.*

- **Line 56: "mass rescue days" this term may not be well understood outside of the rip current community, so I would add a brief definition in parenthesis here.**

Given that it is not further used in the paper, reference to mass rescue day has been removed.

- **Line 131: "resulting in a maximum tide elevation error of 0.3 m at all sites" – how was this error determined?**

This is a (conservative) empirical estimation given that we have no other tide gauge along the coast but only, this is now clarified L136: *... resulting in an estimated maximum tide elevation error of 0.3 m at all sites, which is conservative.*

- **Line 136: Insolation is an uncommon parameter in coastal hazards research (I think) and perhaps warrants a brief definition, along with explanation of how the parameter was recorded or predicted.**

Done L194: *... and hourly insolation I defined as sunshine duration.*

- **Figure 4, upper panel: the symbol for the second legend entry cannot be seen**

Fixed

- **Line 211: you mention model runs, but it is not clear what you mean by 'run' is a run one iteration through the k-fold data divisions? Please clarify this. Also, it is not clear whether the k-folds evaluation was used purely to test the BN or to actually infer the best probabilities within the model; can you clarify the wording in this paragraph to make it more explicit how the model was calibrated/validated.**

Thank you for this comment. A run is indeed one iteration through the k-fold data division. The K-folds evaluation was used purely to test the BN. Best probabilities within the model were fitted differently at each run. We rephrased the L211 as follow: "For a given set of data, a BN fit can have slightly differently results at each run because it is a probabilistic process."

- **4: you state that a value of sk = 1 means perfect prediction, but the equation multiplies by 100% indicating that the values would be from 0-100% with sk = 100% being perfect prediction. Please check this again.**

Thank you for pointing this. 100% has been removed from equation (4)

- **Figure 6 – Major Correction needed. Is this a result or an example? If it is a result, there are a large number of false negatives, which are the worst possible outcome for a predictive model of this kind. This figure makes it look like the model performed very poorly, and therefore the implication of the results in this table needs to be discussed in far more detail than you provide. For instance, out of the 56 observed injuries in the confusion matrix, only 4 of them were correctly identified by the model – is this correct? If so, I would personally throw the model out, but I suspect this is not the case, based on your other skill metrics. Please check this matrix again, and if the results are correct, you need to re-model your incidents in a different way that reduces the occurrence of false negatives.**

This matrix was first presented as an example to provide additional understanding of the ROC curve, but according to comments made by both reviewers it was a misleading piece of information. The confusion matrix depends on the threshold used and cannot be useful as is. Indeed, such a threshold is arbitrary and may vary depending on the goal of the model (do we want a good negative predictive value or a good positive predictive value?). Given that this figure might lead to more confusion than clarification, it was removed in the revised manuscript. All the necessary information for the classification performance is provided by the ROC curve.

- **Line 252: you define variable M, but this variable does not appear in Eqs. 6-8. Is this missing from one of the equations?**

In the first sum of Eq (8) 'N' was replaced by 'M'

- **Section 4.1: you explain more clearly here how the calibration/validation was performed than in the methods section. I would suggest a slight rewording to make it very clear how it was performed – instead of "For calibration, the BN was used to make predictions of an injury based on the input variables of the training cases" it might be clearer to say "For calibration, the BN was trained to make predictions of an injury based on the input variables of 90% of the training cases."**

Done.

- **Line 258: when you mention complexity are you only referring to number of bins in the discretisation of each variable, or the number of parameters included in the BN? Please clarify this.**

We refer to the level of definition, i.e. number of bins, this is now clarified in the revised manuscript.

- **Lines 292-296: this is an interesting finding, but I think needs to be re-worded slightly as on first reading this did not make sense to me. Now I have read it a few times I think I understand what you are trying to say is that when the hazard parent variables are**

**combined within the BN via the latent hazard variable, they make up the equal largest contribution to life risk, but when those variables are considered separately within the BN and instead linked directly to life risk, they individually contribute relatively little to the overall life risk. Therefore those parameters are required to combine/occur simultaneously in order for the model to explain rip current hazard. I would therefore also avoid stating in lines 296-297 that the exposure variables are the most sensitive parameters for the rip current BN; this is misleading as this is only the case when the latent variables are not included (which is not the final BN you presented earlier).**

We agree with the reviewer that it was misleading, we now have reworded so that exposure and hazard exposure are discriminated L300-303: *Within the exposure variables, the most sensitive for the rip-current BN are insolation (Vr = 1.67%), temperature (Vr = 1.7%) and hour of the day (Vr = 1.56%). Within hazard-related variables Hs and T02 have the highest Vr with 0.36% and 0.26%, closely followed by wave direction with 0.25%, suggesting that...*

- **Lines 311-314: these lines at first seem to contradict themselves. I would clarify that by increasing the wave height variable, the latent hazard variable increased and life risk therefore also increased. Because the exposure latent variable is not linked to wave height (nor is exposure observed), it doesn't decrease in your BN when you increase wave height, but you can infer that exposure lowers as wave height increases because wave height is intermediate when highest life risk is predicted. Also, is it not possible that life risk is lower at high wave heights purely because they happen less often in summer, and therefore there are less observations at large wave heights? Please explain in the text whether BNs account for the background distribution in the forcing conditions (e.g. in the way that Scott et al (2014) account for the background distributions).**

These lines have been reworded into L314-319: *Noteworthy, when the BN was updated for larger wave heights, the probability of a shore-break related injury increased. However, when the BN was updated with the evidence of an injury, intermediate wave height bins 0.75-1.5 m and 1.5-2.25 m showed increased probability. This supports our hypothesis that large shore-break waves (Hs > 2.5 m) can discourage bathers from entering the water, which results in less SZIs despite that the shore-break hazard is increased.*

It is true that high wave conditions occur less often in summer. However, when we compared the BN updated for an injury, probabilities of larger wave height bins were lower compared to prior probability of larger wave height. The prior probability BN is based on all daily hours during summers, whether there is an injury or not. Consequently, the comparison of prior and updated BN does take into account all summer conditions

- **Lines 315-318: I think these lines are misleading as they suggest that your IFS variable changes with the tide, when in fact IFS was always observed between 1-3 m amsl. IFS changes in this context are therefore due to gradual morphological changes in beach slope, rather than reflecting the slope of the beach face at different stages of the tidal cycle. Can you please check your logic here, and consider re-wording or clarifying these lines please?**

According to your comment and one of the other reviewer, the part was removed from the revised manuscript.

- **Lines 320-321: It is slightly misleading to compare shore break injuries to rip current 'injuries' in the way you do here – you are actually reporting the rate of rip current drownings (fatal or non-fatal), which is a much more severe 'injury' than many of those in the shore break injuries data and probably occurs less often as a result. They are not easily comparable, as there is a whole spectrum of injuries you could get from a shorebreak, from a sprained ankle to drowning, whereas with rip currents a person either survives (and probably nothing reported) or drowns (equivalent to the most severe shore break injury). Essentially, the rip data is almost binary while the shorebreak data represents a continuum of injury severity. Therefore, that sentence should be reworded to say that a shore break injury is more likely than a rip current drowning (fatal or non-fatal).**

The sentence has been reworded accordingly.

- **In section 4.2 you discuss how the driving parameters affect the likelihood of a shorebreak or rip current related incident, but you do this by assuming in the BN that there is a an incident occurring and looking at which forcing variables increase/decrease. I suggest it would also be useful to explore how the percentage likelihood of an incident changes when you change each forcing variable one at a time (e.g. from max to min). This is somewhat addressed in section 4.1.1, but this doesn't provide insight into how the percentage of incidents changes with each variable, only how the variance of the incidents changes.**

Implicitly we account for this by comparing the sensitivity of each variable to the injury variable. This is in terms of % variance reduction, which is a measure for sensitivity. This gives an idea of the magnitude of influence each individual variable has on the injury variable (specifically on precision of prediction) and which combination of variable interactions are useful to explore. In the scenario analysis section 4.2, we highlighted the most noticeable interactions, indeed by using an injury as evidence. Changing one variable at a time gives, in general, little response to the probability of an injury. This, we think, makes sense, because it is particularly the *interaction* between the evidence of *many* variables, which raises the probability. The evidence of a high tide by itself does not dramatically raise the probability of a shorebreak injury, high tide is not the only explanatory variable given how we constructed the BN. If hypothetically we create a BN with just tide, wave height and sun time, the tide might have a more significant effect on the probability of an injury, but it will not be accurate. Accordingly we did not run additional analysis.

- **Lines 334-338: You have set rip hazard to 100% and explored the change in shoreline sinuosity and wave angle, but rip hazard in your BN is a latent variable that doesn't actually relate to an increase in incidents in the real world. Can you provide more justification on how this latent variable can tell us something about rip hazard/incidents please? Shouldn't you instead vary these forcing variables and observe the change in probability of a rip incident? Or, are you simply interested to see how sinuosity changes as wave angle changes? Please clarify this paragraph.**

Yes, here we were essentially interested in the variable interaction, here between sinuosity and wave

angle. This is now better frames L335-341: *Other rip-current BN scenarios were tested, providing insight into variable interactions. For instance, an interaction between the magnitude of shoreline sinuosity S and wave angle of incidence was explored, given average wave conditions (wave height and period) and a 100\% chance that there was a rip hazard (Figure 12a and b). It can be observed that for a low beach sinuosity (1-1.06) is correlated with higher probabilities for the shore-normally incident waves (around 279]), while large beach sinuosity is associated with a larger probability for more NW angles of incidence (295.43-311.57°). It indicates that for reasonably alongshore-uniform beaches more shore-normally incident wave conditions are required to have a rip current hazard, which is not necessary for rip-channeled beaches.*

- **Line 344: while Stokes et al (2017) did model hazard and exposure variables, they were not hidden variables in their study, as I believe they had observations of number of people in the water (exposure) and incidents divided by the number of people in the water (hazard).**

The reviewer is right, reference to Stokes et al. (2017) has been removed her**e.**

- **Line 359: I would suggest adding a citation to Scott et al. (2014) 'Controls on macrotidal rip current circulation and hazard' as this paper forms the basis of rip predictions used throughout the UK for the lifeguards there, and uses the approach you mention.**

Done

- **Line 380: "dn does not affect the hazard posed" – it would be worth pointing out that tidal currents (not rip currents), which are driven by dn, are typically of very low velocity compared to rip currents, which is why you can state that dn is not a driver of hazard directly, while n itself is a driver of hazard as it determines whether the surf-zone is located around, and therefore interacts with, the bar-rip features.**

This part of the discussion has been rewritten and now reads L380-385: *We also found that rapid, positive or negative, change in tide level elevation (large |d\eta|) increase the probability of drowning incidents, with no difference between ebb and flood. Given that tide-driven current are negligible compared with rip currents along most of the beaches in southwest France, this suggests that rapid changes in tidal elevation driving the rapid onset of rip current activity can surprise unsuspecting bathers and carry them offshore. However, another explanation is that some of the drowning incidents occurred in sectors adjacent to the Arcachon lagoon and Gironde estuary where tide-driven currents, which are maximized during ebb and flood (large |d\eta|), can be intense.*

- **Line 390: it is not clear how you came to this conclusion. Can you please clarify this finding earlier on in the paper. I did not come to the same conclusion by reading lines 334-338, which is what I think you are referring to.**

We agree, according to our response to a previous comment, this now reads L391-396: *Our scenario analysis also indicates that, for reasonably alongshore-uniform beaches, more shore-normally incident wave conditions are required to have a rip current hazard compared with rip-channeled beaches. This is also in line with observations and model outputs showing that, for the same obliquely incident wave conditions, rip cell circulation are transformed into an undulating, less hazardous,*

*longshore current for weakly (small S) alongshore variable surf zone morphology, while rip cell circulation can be sustained for deep rip channels (MacMahan et al., 2016; Dalrymple et al., 2011).*

- **The role the latent variables play in your BN warrants further explanation. Unlike previous studies that have used hazard and exposure in beach life risk studies (for example, Stokes et al., 2017 who you cite) these variables in your BN are not observed variables. Therefore, please explain further how the BN estimates an increase in exposure or hazard based only on the forcing variables and the resulting change in number of incidents.**

The latent variable uses correlations among child nodes to determine the relation between the predictor variable (injury). The latent /hidden node should be interpreted as the (discretized) probability distribution of these correlations amongst child nodes in relation to the predictor variable (injury). This is frequently used when there is an unobserved cause. The latent variables in our BN are the representation of our beliefs translated into probabilities. The goal of this is to show how important these variables are and how they change, given a results (SZI) and/or an environmental parameter.

- **Your model performs poorly when measured with LLR and to some degree 'skill', yet seems to perform very well against AUC. It is therefore hard to place confidence that the model is useful, when some metrics say it performs worse than prior probability. Can you comment on which skill metric is the most appropriate to measure your BN against as it is hard otherwise to conclude on its usefulness.**

Skill and summed LLR are the lesser metrics, because skill does not take into account the confidence of the model in its prediction. The summed LLR is sensitive to highly negative 'anomalous cases' that are hard to predict, giving a distorted view of the model performance. The %LLR>0 nuances this distorted view. The AUC/ROC does take into account the confidence of the model to some extent with a cut-off value, and is therefore better for binary metrics.

**Technical corrections**

- **Abstract line 6: "a hidden hazard and exposure variables" should read "hidden hazard and exposure variables"**

Done

- **Introduction line 57: "…the number of drowning incidents occur during warm sunny days…" should read "…the number of drowning incidents increases disproportionately during warm sunny days…"**

Done

- **Introduction line 63: "the benefits of a BN approach to identify of the characteristics of high risk beaches from a large data set." Should read "the benefits of a BN approach to identify the characteristics of high risk beaches from a large data set."**

Done

- **Line 76: "The Gironde coast is located in southwest of France" should read "The Gironde coast is located in the southwest of France" or "The Gironde coast is located in**

**southwest France"**

Corrected

- **Line 139: "beach upper beach slope" should read "upper beach slope"**

Corrected

- **Line 324: reword to 'rip current related drownings are slightly more likely to occur when tides are low' or similar, as cause and effect are the wrong way round in your sentence.**

Done

- **Line 325: "such drowning incidents occur for increased incident wave energy" should read "such drowning incidents occur during increased incident wave energy"**

This sentence has been modified according to a comment by the other reviewer and now read L325-326: *Rip current related drownings are slightly more likely to occur when tides are low, with increased probabilities for larger Hs and T02.*

- **Line 341: please re-word this sentence as it does not make sense.**

This introduction sentence was not crucial and has been removed.

- **Line 342: "This allowed to use different beach" should read "Thisf allowed the use of different beach"**

Done

- **Line 345: It is not clear why the word 'importantly' is used in this sentence/line.**

"Importantly" has been removed.

- **Line 355: "Summer beach profiles acquired along the different beaches should help improving IFS estimation and, in turn, prediction of rip-current drowning incidents." I think this is incorrect, as you don't relate IFS to rip incidents, only S.**

This sentence has been rewritten into L357-358: S*ummer beach morphologies surveyed along different beaches distributed along the coast should help improving IFS and S estimation and, in turn, prediction of rip-current drowning incidents and shore-break related injuries.*

- **Line 361: "the risk of drowning the Gironde" should read "the risk of drowning along the Gironde coast"**

Done

- **Line 363: "could not be retrieved to either" should read "could not be attributed to either"**

Done

- **Line 394: "including a wisely pre-defined hidden hazard variable" this sounds a bit self-congratulating! I would remove the word 'wisely'**

Done

- **Line 399: "with decreased exposure for Hs > 2.5 m, large surf, and thus heavy shore-break waves at the shoreline, discourage the beachgoers to enter the water near high tide." reword to "the predicted decrease in exposure for Hs > 2.5 m, representing heavy shore-break waves at the shoreline, is thought to discourage beachgoers from entering the water near high tide"**

Done

- **Line 404: "more controlled by the exposure than by hazard" reword to "more controlled by exposure related variables than by hazard related variables"**

Done

- **End of line 411: 'improve' rather than 'improves'**

Done

- **Line 420 "skill but providing much less diagnostic Tellier et al. (in revision)." Should read "skill but providing much less diagnostic capability (Tellier et al., in revision)."**

Done
* * *
**REVIEWER #2**

**Reviewer comments**

Our response

*Changes in the revised manuscript*
* * *
**Authors analyse two beach safety-related hazards in the Atlantic French coast and propose a BN-based model to describe/predict them as a function of a hazard and an exposure component, each one comprising relevant variables characterising environmental (hydrodynamic and climatic/weather) conditions. The topic is in line with one of the targets of NHESS and, in this sense, the manuscript can be of interest for many NHESS readers. In what follows, some observations/comments/suggestions are given.**

We thank the reviewer for their insightful comments and support for publication. Below you will see that all the comments have been carefully considered and that, when possible, the required changes have been made.

**General comments**

**[1] The proposed model includes different environmental variables to characterize the hazard and exposure components. With respect to the exposure component, the implicit hypothesis is that the meteorological conditions together the time of occurrence should indicate the beachgoers affluence. Although it seems a reasonable assumption, it would be recommendable to perform an independent validation because the errors/uncertainty in this "submodel" (weather conditions controlling number of beachgoers) should affect the predictability skill of the overall model (dependence of the probability of SZI on number of beachgoers). In a previous paper (Castelle et al 2019), the authors concluded the need of better quantify exposure to better explain SZI. Is there any curve of affluence for the studied beaches which can be used to "validate" the conceptual model of exposure?**

We agree with the reviewer that beachgoer affluence, and particularly exposure (affluence of beachgoers in the surf), should be validated. Unfortunately, at the time of writing this response we still have no affluence / exposure data. We have a proposal currently under review which one of the aims will be to estimate beach affluence, including number of people in th bathing zone at the time of each accident in the future injury report forms that are filled by the lifeguards. In line with Castelle et al. (2019) we further indicate that research effort must be made into beachgoer affluence and exposure to hazard in the surf L407-408: *This will also involve estimation of beachgoer affluence, and estimation of the number of people in the surf exposing themselves to the physical hazards.*

**[2] Line 137. Please define "tidal gradient".**

Tidal gradient was defined as the time derivative of tidal elevation, this is now specified in the revised manuscript when introducing the variables.

**[3] Line 148. Please specify the number of the "Figure".**

Done

**[4] Figures 3 and 4 can easily be combined in just one.**

We think that, for readability and to better discriminate shore-break and rip-current related injuries, it is better to keep two different figures.

**[5] Lines 196-198. One of the implications of discretizing injuries in binary form (and duplicating cases when exceeding 1) is to artificially augment data to be used in the BN. May this affect the real representativeness of environmental conditions affecting SZI? Also, this may also artificially increase the BN predictive performance.**

The objective is indeed to artificially increase the amount of data to be used in the BN. It is similar to a weight function. It is a common way to improve prediction model performance (BN or other models). It will not falsely increase performance as long as the model is tested/validated against different dataset (i.e. a replicated data are not splitted between fit and validation datasets).

**[6] Figure 6.b. The confusion matrix indicates a poor performance of the model to predict injuries. I assume that this matrix corresponds to one of the tested fold runs (in fact, one with the poorest performances). However, as it is presented it seems that this is the overall performance of the model, which if so, it indicates the absence of reasonable predictive model. Please put the matrix in the right context.**

The Reviewer is right. According to a comment by the other reviewer, this panel has been removed as it provided misleading information.

**[7] You propose three different metrics to measure the BN predictive performance. Do we really need to use all of them to identify the best number of bins to be used in the model? If so, how their results must be jointly interpreted (integrated)?**

See response on RC1. Metrics are complementary. Skill is less suitable for binary predictors and does not take into account the confidence of the model, whereas LLR does this. AUC/ROC takes the ratio of false positives and true positives, which is a natural for a binary metric. However, this does also not take into account the confidence of the model

**[8] Line 252. M is defined, but where do you use it?**

In the first sum of Eq (8) 'N' was replaced by 'M'

**[9] Line 278-279. So, finally, how many bins are selected for each BN. Looking to the final BNs (figures 10 & 12), you have selected 6 for shore-break and 7 for rip-current. However, the variation in prediction for both when changing from 6 to 7 is almost the same. Why didn´t you use the same number of bins in both BNs? This should give more consistency to the analysis since input variables which are almost the same will be discretized in the same way.**

Given, that the AUC/ROC does perform significantly better for the 7 bins, we chose 7 bins. Even though for Skill/ Summed LLR variations are small. Choosing the same number of bins for both BNs would result in at least one sub-optimal model, which we avoided.

**[10] Line 258. Are you using "complexity" to refer to the number of bins used. I would not say this is complexity but "level of definition" (or similar). I understand "complexity" by the number of components/variables used in the model.**

The reviewer is right, this sentence now reads L262-263: *Generally an increase in level of definition, i.e. number of bins, leads to a decrease in predictive capability and vice versa ...*

**[11] Line 289. You mention that beach profiles were taken only at one location. Unless that the beach is alongshore uniform, this may significantly affect to the role of IFS within the BN. Since you are using sinuosity (to predict rip SZI) as a measure of beach departure from alongshore uniformity, I presume that the slope will change along the beach. Do you have an estimation of the range of variation of IFS? Maybe this is the reason for the very low contribution of IFS to the variance.**

The writing was misleading here, thank you for pointing out this issue. As indicated in Section 2.2 and shown in Figure 3a, in fact four profiles distributed along the coast were used to account for the alongshore variable beach slope. This sentence has been rewritten into L293-294: *The inverse foreshore slope IFS averaged over four profiles distributed along the coast has a noticeable impact (Vr = 0.065%) on the shore-break hazard.*

**[12] Line 293. I think that figures referring % variance of Exposure and Hazard are wrong (figure 9c). Exposure is the larger than hazard.**

Thank you for pointing this. This has now been fixed.

**[13] Figure 9. Can you use the same scale in the X-axis for both hazards for an easier visual comparison? Are figures 9b and 9d just a zoom of figures 9a & 9c after removing the first two contributions (hidden variables)? If yes, please indicate, as it is written it suggests they were obtained independently.**

We stress that the variance reduction is model dependent and should not be interpreted as an absolute metric to do inter-model comparison. Therefore, for better clarity of the figure we kept the same scale as in the former manuscript. Figures 9b and 9d are not just zooms.

**[14] Lines 294-301. When you compare the contribution of hazard latent variables to the variance for shore-break and rip-current hazards, you obtain a larger contribution in the case of rip than for shore-break. May this indicate that the built model (selected variables) for the shore-break is worse than the one for rips? (e.g. the above-mentioned potential effect of neglecting variations in IFS -comment [10]-)**

This should not be interpreted as a inter-model comparable performance metric, but as a model dependent sensitivity to the target variable (injury). Consequently, we can only conclude that hazard is less sensitive to injury then exposure is for the shorebreak BN and vice versa for the rip BN.

**Line 297. If we compare the %variance of exposure variables (hour, temp, I) in both hazards (fig 9b & 9d), they are very similar (different ordering but same order of magnitude).**

We agree, this is now indicated L304-305: *Interestingly enough, the percentage of variance reduction of exposure variables are of similar magnitude, although with different ordering, in both hazards (Figures 9b,d).*

**Line 300. Formally, you are not including wave energy as a variable in your BN. Consider that wave energy will involve a non-linear combination of H and T and the observed contribution of these individual variables may significantly vary when combined to characterize wave energy.**

The reviewer is right, this has been rewritten into L???: *Hs and T02 have the highest Vr with 0.36% and 0.26%, closely followed by wave direction with 0.25%, suggesting that incident wave conditions are the most important control on rip-current hazard.*

**[15] Line 311. Where the update for larger wave heights (and the resulting change in % of shore-break injuries) can be seen?**

It was not shown in order to reduce the number of figures, this is now clarified L304-305: *Noteworthy, when the BN was updated for larger wave heights (not shown), the probability of a shore-break related injury increased.*

**[16] Lines 315-318. I would say here that the most likely IFS during a shore-break hazard is a steep slope whatever the tidal elevation is. Then, if you want you can highlight secondary differences in IFS, but in any case you are just concentrating in one single class for intermediate IFS (27.5 to 35), but when you refer to intermediate you could also include (20 to 27.5) and then, the probability at both tide levels would be almost the same. Furthermore, trying to draw any conclusions about IFS at low tide does not seem to make much sense, since IFS is measured above mean sea level.**

We agree. This paragraph and Figure 10 have been removed.

**[17] Lines 321-323. What are larger tidal gradients? If we consider the three central bins as representative of medium-low gradients, they are concentrating a similar % of occurrence. In fact, if we compare the probability distribution is almost similar to the prior one (Fig 11a).**

The reviewer is right, differences are slight and mostly for the extreme tidal gradients, this is now clarified L322-323 : *Although the probability distribution of the central bins of tidal gradients are similar in pattern, extreme tidal gradients ($|d\eta| > 0.43$ m/hr) show an increase in probability by c. 50%.* The discussion on tidal gradients has been changed accordingly L380-385: *We also found that rapid, positive or negative, change in tide level elevation (large $|d\eta|$) increase the probability of drowning incidents, with no difference between ebb and flood. Given that tide-driven current are negligible compared with rip currents along most of the beaches in southwest France, this suggests that rapid changes in tidal elevation driving the rapid onset of rip current activity can surprise unsuspecting bathers and carry them offshore. However, another explanation is that some of the drowning incidents occurred in sectors adjacent to the Arcachon lagoon and Gironde estuary where tide-driven currents, which are maximized during ebb and flood (large $|d\eta|$), can be intense.*

**[18] Lines 326-328. It is not clear from Figure 12b that more sinuous shorelines show increased probability of injuries. The updated distribution of Sin is quite similar to the prior distribution. Are the small changes detected large enough to support your conclusion?**

We agree that the difference is small, we slightly reworded this part L327-329: *More sinuous shorelines (larger S values) show slightly increased probability of rip-current related drowning (see for S > 1.23 in Figure 11b), suggesting that more alongshore-variable surf zone morphology increases the rip-current hazard.*

**[19] Lines 329-332. I disagree that your analysis is really reflecting that the peak of rip injuries (13h – 15 h) is much earlier than the one for shore-break one (14 to 16.33). Both bins overlap, which may be associated with the comparison of different bins resulting from using 6 classes in shore-break and 7 classes in rip for a same variable (Time). According to Castelle et al (2019) "For low TR, daily minimum tide elevation, which is when channel rip activity is maximised, tends to occur during the patrolled hours in the mid-to-late afternoon (Fig. 12b) when beach attendance (exposure) is maximised". This does not seem to fully support your conclusion.**

For the first point made, we agree that "much earlier" was too strong a statement, it has been reworded into L331-332: *Furthermore, the highest peak is earlier between 13h-15h (Figure 11b) compared to shore-break injuries, although the bins slightly overlap (14h-16.33h, Figure 10b).* We think that the comparison with Castelle et al. (2019) is not relevant here as it deals with low TR only, while here all tide ranges are considered.

**[20] Legend of Figure 13. Please change the legend to something similar to the one used in Fig 11 (e.g. Scenario with low sinuosity resulting in a higher probability of shorenormal wave direction). As it stands, it seems that you build the scenario by fixing booth sinuosity and direction.**

Figure 2 caption now reads: *(a) Scenario with low sinuosity resulting in more shore-normal wave direction (around 279°) with the rip current BN; (b) Scenario with medium sinuosity resulting in more obliquely incident (NW) wave conditions with the rip current BN.*

**[21] Lines 347-348. I agree with this comment. It would be interesting to assess the profile of people injured by shore-break and rips to identify potential factors affecting their relative exposure.**

Thank for this comment. It is, together with beachgoer attendance, the topic of a research proposal with social scientists which is currently under review.

**[22] Lines 359-360. See also the combination of video images and numerical modelling to help**

**managing beach safety (Jiménez et al. 2007. Beach recreation planning using video-derived coastal state indicators. Coastal Engineering, 54, 507-521).**

Thank you, reference to this site is now included L363-364: *...or on the combination of video images and numerical modelling (Jimenez et al., 2007).*

**[23] Section 5.2. Please adjust comments on the role of different environmental factors according to your response to previous comments [e.g. 16, 17, 18, 19]**

This section has been rewritten in depth according to your comments and those by the other reviewer.